**JCB** Journal of Cell Biology

## REPORT

# Unraveling the kinetochore nanostructure in *Schizosaccharomyces pombe* using multi-color SMLM imaging

David Virant[1]*, Ilijana Vojnovic[1,2,3]*, Jannik Winkelmeier[1,2,3]*, Marc Endesfelder[4], Bartosz Turkowyd[1,2,3], David Lando[5], and Ulrike Endesfelder[1,2,3]

The key to ensuring proper chromosome segregation during mitosis is the kinetochore (KT), a tightly regulated multiprotein complex that links the centromeric chromatin to the spindle microtubules and as such leads the segregation process. Understanding its architecture, function, and regulation is therefore essential. However, due to its complexity and dynamics, only its individual subcomplexes could be studied in structural detail so far. In this study, we construct a nanometer-precise in situ map of the human-like regional KT of *Schizosaccharomyces pombe* using multi-color single-molecule localization microscopy. We measure each protein of interest (POI) in conjunction with two references, cnp1[CENP-A] at the centromere and sad1 at the spindle pole. This allows us to determine cell cycle and mitotic plane, and to visualize individual centromere regions separately. We determine protein distances within the complex using Bayesian inference, establish the stoichiometry of each POI and, consequently, build an in situ KT model with unprecedented precision, providing new insights into the architecture.

## Introduction

The proper segregation of chromosomal DNA during cell division is one of the most crucial processes in the cell cycle of living organisms. Aneuploidy, caused by chromosome maldistribution, leads to cancer, birth defects, and cell death (Pfau and Amon, 2012; Santaguida and Amon, 2015; Yuen et al., 2005). During chromosome segregation, sister chromatids are separated by the microtubules of the spindle apparatus. Here, the kinetochore (KT), a multiprotein complex, acts as the link between the centromeric DNA and the microtubules emerging from the spindle (Musacchio and Desai, 2017; Roy et al., 2013).

In budding yeast *Saccharomyces cerevisiae*, the KT appears as a structure of about 126 nm in diameter (Gonen et al., 2012). A single KT connects one single microtubule to a 125 base-pair region, a so-called point centromere of defined sequence (Clarke, 1998; Winey et al., 1995). In contrast, in the fission yeast *Schizosaccharomyces pombe*, the KT complex links 2–4 KT microtubules (kMTs) to a regional centromere in the kilobase-pair range (Chikashige et al., 1989; Ding et al., 1993). In higher eukaryotes, the structure is even more extensive. For human chromosomes, the centromeric region spans 0.5–1.5 megabases (Wevrick and Willard, 1989; Zinkowski et al., 1991). Accordingly, the human KT structure connects to about 20 kMTs (McEwen et al., 1998; McEwen et al., 1997).

Despite these differences in centromere length and number of kMTs, the general KT architecture is highly conserved (Drinnenberg et al., 2016; van Hooff et al., 2017). The centromeric region is epigenetically defined by a H3 histone variant, cnp1[CENP-A], which replaces one or both H3 in the centromeric nucleosomes (Dunleavy et al., 2011; Lando et al., 2012). (In this report, we use the terminology of *S. pombe* and the human homolog in superscript, if different.) cnp1[CENP-A] provides the scaffolding for the KT, which consists of several subcomplexes: the CCAN network, the MIS12/MIND complex (MIS12c/MINDc), the KNL1 complex (KNL1c), the NDC80 complex (NDC80c), and the DAM1/DASH complex (DAM1/DASHc or Ska complex in humans; Kixmoeller et al., 2020; Musacchio and Desai, 2017).

The structures of these subcomplexes have been made available by cryo-EM and x-ray crystallography, e.g., of the CENH3 nucleosome (Migl et al., 2020; Tachiwana et al., 2011), the CCAN network (Hinshaw and Harrison, 2019a; Hinshaw et al., 2019b; Yan et al., 2019), the MIND/MIS12c (Dimitrova et al., 2016; Petrovic et al., 2016), the NDC80c (Ciferri et al.,

[1]Department of Systems and Synthetic Microbiology, Max Planck Institute for Terrestrial Microbiology and LOEWE Center for Synthetic Microbiology, Marburg, Germany; [2]Department of Physics, Carnegie Mellon University, Pittsburgh, PA, USA; [3]Institute for Microbiology and Biotechnology, Rheinische-Friedrich-Wilhelms-Universität Bonn, Bonn, Germany; [4]Institute for Assyriology and Hittitology, Ludwig-Maximilians-Universität München, München, Germany; [5]Department of Biochemistry, University of Cambridge, Cambridge, UK.

*D. Virant, I. Vojnovic, and J. Winkelmeier contributed equally to this paper. Correspondence to Ulrike Endesfelder: endesfelder@uni-bonn.de.

2008; Valverde et al., 2016), and the DAM1/DASHc (Jenni and Harrison, 2018). Despite advances in reconstituting large subcomplexes, e.g., resolving the 13-subunit Ctf19c/CCAN (Hinshaw and Harrison, 2019a), the assignment of KT proteins to specific subcomplexes within the fully assembled KT has not been successful so far (Gonen et al., 2012; Walstein et al., 2021).

Fluorescence microscopy visualizes proteins of interest (POIs) using fluorescent labels such as fluorescent proteins (FPs; Tsien, 1998). Quantitative fluorescence measurements provide KT protein copy numbers (Coffman et al., 2011; Dhatchinamoorthy et al., 2017; Joglekar et al., 2008; Joglekar et al., 2006; Johnston et al., 2010; Lawrimore et al., 2011; Schittenhelm et al., 2010; Suzuki et al., 2015) and intra-complex distances (Aravamudhan et al., 2014; Haase et al., 2013; Joglekar et al., 2009; Schittenhelm et al., 2007; Suzuki et al., 2014; Suzuki et al., 2018; Wan et al., 2009). However, the spatial resolution is not sufficient to resolve individual KTs (Tournier et al., 2004). Here, super-resolution microscopy bridges the resolution gap between conventional fluorescence microscopy and electron microscopy. The KT has been studied by Structured Illumination Microscopy (Dhatchinamoorthy et al., 2019; Schubert et al., 2020; Venkei et al., 2012; Wynne and Funabiki, 2016; Zielinska et al., 2019) and Stimulated Emission Depletion microscopy (Drpic et al., 2018; Zielinska et al., 2019). Both provided detailed pictures of the structure at about 100 nm resolution. Only a handful of studies have been carried out at single-molecule resolution, so far. Two in situ studies each targeted a single protein, cnp1[CENP-A] (Lando et al., 2012) and Ndc80 (Wynne and Funabiki, 2016), and one study focused on two-color pairs in vitro (Ribeiro et al., 2010).

In this study, we investigate the architecture of the regional KT of the fission yeast *S. pombe*. We created a strain library of fluorescent fusions of key proteins of the inner and outer KT. Using quantitative multi-color single-molecule localization microscopy (SMLM) and Bayesian inference, we report in situ inner-KT distances and protein copy numbers. Based on those numbers, we can determine the KT architecture during mitosis and build an in situ model at a nanometer resolution. Our study confirms in vitro studies on reconstituted KT subcomplexes and adds new *S. pombe* specific insights to our current KT knowledge.

## Results and discussion
### A strategy to unravel the KT structure
To investigate the KT architecture during mitosis, we first designed a robust and quantitative multi-color SMLM imaging strategy, which did not exist before (Vojnovic et al., 2019). Specifically, our aim was to measure protein stoichiometries and distances between different POIs to build a molecular KT map (Fig. 1, A and B). We set up a triple-color (3C) strain library of two labeled reference proteins and a varying POI: A spindle reference to identify cells in metaphase/early anaphase and the orientation of the mitotic plane and a second, centromeric reference as the landmark for KT assembly. To be a reliable reference, the chosen proteins need a defined organization, sufficient abundance, and to be present throughout the cell

cycle. This led us to choose sad1, a protein from the spindle pole body (SPB; Vojnovic, 2016) and the centromere-specific histone cnp1[CENP-A] (Lando et al., 2012).

From our imaging priority (1. registering the mitotic plane, 2. quantitative readout of POI stoichiometry and distribution, 3. quantitative readout of cnp1[CENP-A] as reference for stoichiometry and for distances), experience on autofluorescence and labeling (Fig. S1), we derived alternative imaging strategies (Materials and methods) and finally decided on using dual-color Photoactivated Localization Microscopy (PALM) imaging by UV- and primed photoconversion (Turkowyd et al., 2017; Virant et al., 2017) together with a diffraction-limited SPB reference as third color (Fig. 1 C). For the FP labels, we chose mScarlet-I (Bindels et al., 2017) as SPB reference and UV-photoactivatable PAmCherry1 (Subach et al., 2009) as centromeric reference and created a dual-color reference strain. For the POIs, we chose primed photoconvertible mEos3.2-A69T (Turkowyd et al., 2017), whose readout we calibrated using a FtnA 24mer standard (Fig. 2, B and C) and built, using the dual-color strain as a template, a 3C strain library targeting in total 10 different key POIs of the inner and outer KT at their C-terminus (Fig. 1 C, Fig. S2, and Table S1).

### Health assessment of the strain library
The integration of FPs may reduce protein functionality and alter cell physiology. Performing spot tests, we checked for temperature sensitivity (25, 32, and 37°C) and for KT-MT attachment defects using thiabendazole (TBZ, 2.5, 5.0, 7.5, and 10.0 mg/ml), an MT depolymerizing drug (Tang et al., 2013). Through flow cytometry, we tested all strains for a WT-like phenotype. The 3C strains showed no deviations from the parental h+/WT strain except for two strains, cnp3[CENP-C]-3C and dam1-3C (Fig. S2, C–E). cnp3[CENP-C]-3C exhibited both a TBZ sensitivity and a notably larger cell size and was excluded from further analyses. Interestingly, cnp3[CENP-C] interacts with the N-terminal CENP-A Targeting Domain region of cnp1[CENP-A] via its C-terminus (Black et al., 2007; Carroll et al., 2010; Guse et al., 2011). Thus, FP tagging of both, the cnp1[CENP-A] N-terminus and cnp3[CENP-C] C-terminus might have caused the observed effects. This hypothesis is supported by measurements using a 1C strain where only the cnp3[CENP-C] C-terminus was tagged, as cnp3[CENP-C]-1C did not show any modified phenotype (data not shown). dam1-3C appeared to be more stable in TBZ spot tests. Interestingly, similar results were obtained before: a dam1 deletion mutant was hypersensitive to TBZ, whereas a C-terminal truncation (dam1-127) led to higher TBZ resistance (Sanchez-Perez et al., 2005). dam1 with two C-terminal mutations also proved to be more resistant to TBZ (Griffiths et al., 2008). This suggests that the dam1 C-terminus is important for controlling kMT stability, which might increase by C-terminal tags. As dam1-3C showed no mitotic delay or altered phenotype, it was included in the study.

### Determining KT distances and protein stoichiometry
All 3C strains were prepared and imaged by a strict protocol that was repeated on different days using biological replicates (Materials and methods). Since single centromeres can only be

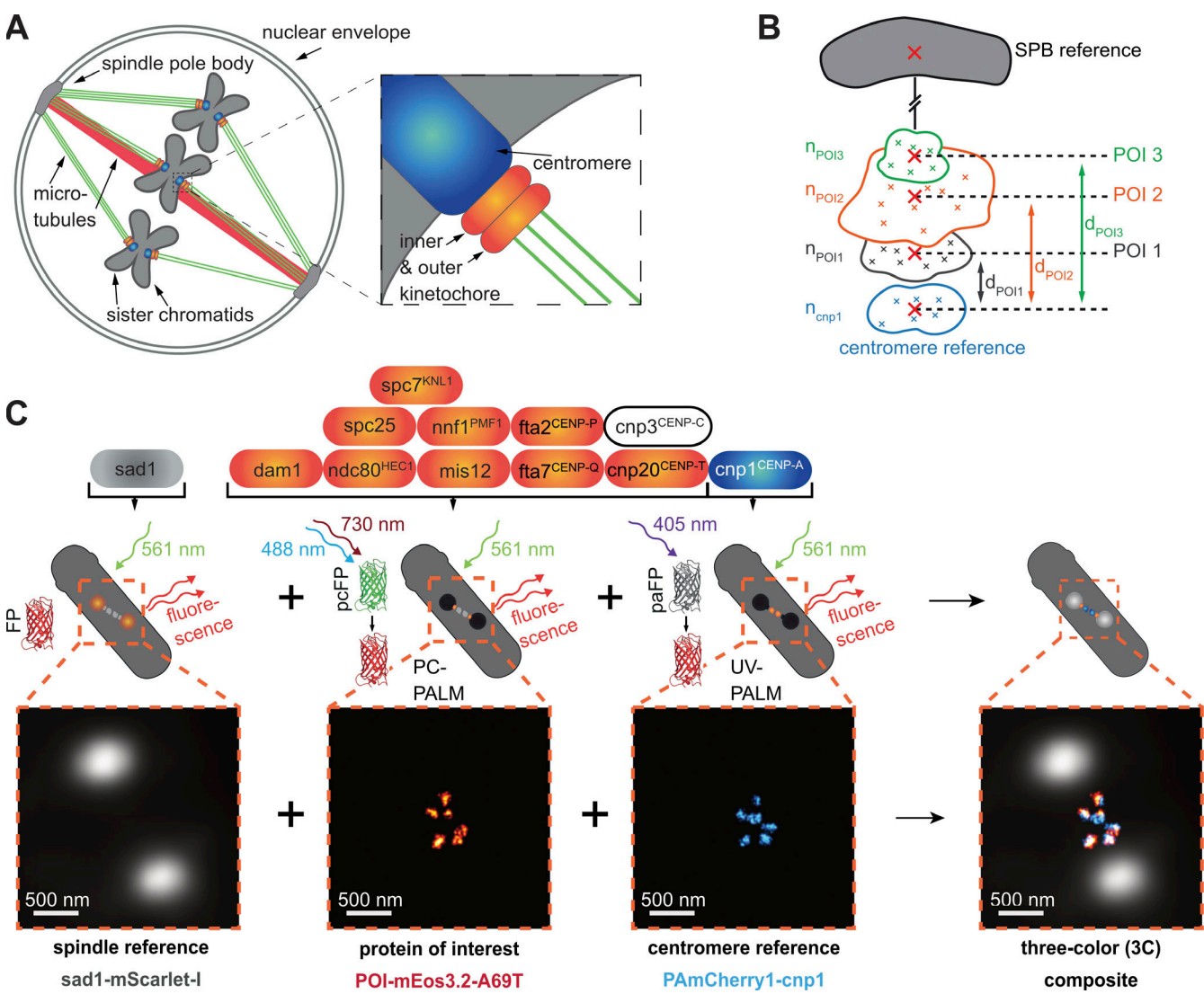

Figure 1. **Imaging strategy for measuring the KT nanostructure. (A)** During mitosis, the three sister chromatid pairs in *S. pombe* are attached to the two SPBs by tethering microtubules (green), and the SPBs are pushed apart by a bundle of spindle microtubules (red). The attachment between the centromeric region of each chromatid (blue) and the microtubules is facilitated by the KT-protein complex (orange), which consists of an inner and outer part. **(B)** Each POI in the KT complex can be mapped into the KT nanostructure by localizing it in relation to two references, a centromere and an SPB reference. Information about its copy number $n_{POI}$ and its orientation and distance $d_{POI}$ to the reference proteins is obtained. **(C)** Imaging strategy: First, the SPB protein sad1-mScarlet-I fusion is imaged by conventional epifluorescence microscopy to map the mitotic spindle. Then, both the POI and the centromere reference cnp1$^{CENP-A}$ are measured by super-resolution microscopy: Here, each POI-mEos3.2-A69T fusion (here spc7) is imaged by primed PC-PALM, followed by a readout of PAmCherry1-cnp1$^{CENP-A}$ fusion using UV-activation PALM (UV-PALM). Scale bar, 500 nm.

resolved during metaphase/early anaphase (Fig. S3; Tournier et al., 2004), the fraction of cells in this phase was increased by cell-cycle synchronization (Fig. S2). All SMLM data were post-processed and annotated by several manual and automated analysis steps and stored in an SQL database (Fig. 2 A). For this work, we focused on (i) the distances of the POIs to the centromeric region marked by cnp1$^{CENP-A}$ and (ii) their copy numbers. For the distances, we used Bayesian inference and derived the posterior probability distribution for each POI-cnp1$^{CENP-A}$ distance (Fig. 2, F and G; Materials and methods). In our Bayesian model (Fig. 2 F), we explicitly accounted for the angular offset between the kMT and spindle axes (Fig. 2 F and Fig. S3). We calibrated the measured localization counts per POI cluster

using bacterial ferritin FtnA, an established homo-oligomer protein standard consisting of 24 subunits (Fig. 2, B and C).

In Fig. 3, we summarized all POI distances to cnp1$^{CENP-A}$ and stoichiometries (statistics in Tables 1 and 2). Our in situ measurements are consistent with work based on conventional fluorescence microscopy (these studies, however, show considerable variation between them) and very accurately confirm structural data from in vitro EM and crystallography considering that (i) typically tens to hundreds of amino acid long sequences of termini are unstructured and therefore missing from structural data, which (ii) also do not carry FP tags of 2–3 nm barrel size. We will discuss the KT structure in the order of inner to outer KT POIs below.

fta2[CENP-P] and fta7[CENP-Q] are part of the COMAc/CCAN network in a strict 1:1 stoichiometry. Their C-termini are close to each other, oriented in the same direction (Hinshaw and Harrison, 2019a; Pesenti et al., 2022). Our in situ distances of 11.4 ± 1.9 nm for fta2[CENP-P] and 11.7 ± 2.9 nm for fta7[CENP-Q] and the copy numbers of 55.1 ± 3.3 and 57.4 ± 4.1 reflect these properties and are consistent with cryo-EM data from reconstituted *S. cerevisiae* COMAc (Yan et al., 2019), with distances of N-cnp1[CENP-A] to fta2[CENP-P]-C and to fta7[CENP-Q]-C being 9.7 and 10.7 nm, respectively.

mis12 and nnf1[PMF1] are both part of the MINDc in a 1:1 ratio (Maskell et al., 2010; Petrovic et al., 2010). Our distances are 28.0 ± 1.6 nm for mis12 and 27.1 ± 1.4 nm for nnf1[PMF1]. They confirm fluorescence data where all C-termini of the MINDc proteins line up (Joglekar et al., 2009) and show Fluorescence Resonance Energy Transfer proximity signal (Aravamudhan et al., 2014) as well as EM data positioning all C-termini near each other (Petrovic et al., 2014). For spc7[KNL1], we found 25.9 ± 1.2 nm. Whereas full-length spc7[KNL1] has not been purified (Maskell et al., 2010), its C-terminus has been resolved by EM (Petrovic et al., 2014) and to be close to the C-termini of mis12 and nnf1 (Petrovic et al., 2016). In fluorescence microscopy, the spc7[KNL1] C-terminus colocalized with the spc24 C-terminus within the NDC80c (reflected by our spc25 data below; Joglekar et al., 2009). Interestingly, in our hands, the MINDc and spc7[KNL1] proteins' stoichiometry amounts to 1:1.3:1.3 (copy numbers of 73.1 ± 3.1 mis12: 92.6 ± 3.6 nnf1[PMF1]: 95.4 ± 3.3 spc7[KNL1]). This differs from the reported 1:1 ratio for mis12: nnf1[PMF1] but posits a 1:1 ratio for nnf1[PMF1] and spc7[KNL1]. From reconstitution and cross-linking experiments, it is known that

numbers and robustness of the method. Localizations per POI cluster. POIs from the same subcomplex are shown in the same colors (COMAc [green]: fta2[CENP-P], fta7[CENP-Q]; MINDc [blue]: mis12, nnf1[PMF1]; NDC80c [red]: spc25, ndc80[HEC1]). The red dot indicates the mean, the green line indicates the median, the black box indicates the SEM, and the whiskers indicate the STD. N = 57 for cnp20[CENP-T], 87 for fta2[CENP-P], 61 for fta7[CENP-Q], 86 for spc7[KNL1], 136 for mis12, 217 for nnf1[PMF1], 86 for spc25, 172 for ndc80[HEC1], 232 for dam1. **(E)** Distributions of localizations per cnp1[CENP-A] cluster are robust across different POI-3C measurements. The gray dotted line indicates the mean of all cnp1[CENP-A] clusters for reference. N = 55 for cnp20[CENP-T]-3C, 96 for fta2[CENP-P]-3C, 68 for fta7[CENP-Q]-3C, 239 for spc7[KNL1]-3C, 197 for nnf1[PMF1]-3C, 122 for mis12-3C, 68 for spc25-3C, 161 for ndc80[HEC1]-3C, 239 for dam1-3C. **(F)** Bayesian model to estimate inner-KT distances. Schematic of our Bayesian model. To determine the real distance (marked in red) between cnp1[CENP-A] and each POI from the measured centers of their respective clusters, we assumed Gaussian measurement errors of uncertain size (dotted gray circles, right). To be able to disentangle the contribution of errors and real distance in the measured data, we took the position of the associated spindle pole into account, as the centroids of sad1, POI, and cnp1[CENP-A] clusters can be assumed to lie on a straight line (kMT axis, left). The sad1 cluster closest to a KT is not necessarily the pole to which the KT is attached to. Thus, we built a mixture model to take both possibilities into account. For each KT pair, we thus obtained two options to check, marked by the green and orange triangle, right. **(G)** The posterior density of cnp1[CENP-A]-POI distances for each POI measured in this study was approximated using Hamiltonian Monte Carlo (see Materials and methods). Number of centromeres used for distance measurement: N = 49 for cnp20[CENP-T], 82 for fta2[CENP-P], 58 for fta7[CENP-Q], 215 for spc7[KNL1], 161 for nnf1[PMF1], 102 for mis12, 51 for spc25, 135 for ndc80[HEC1], 155 for dam1. The code can be found in Data S1.

Figure 2. **Data analysis. (A)** Schematic representation of the data analysis pipeline. In the image processing part, image data from SMLM experiments are localized and post-processed (for quality, drift, etc.). The resulting localization tables are added to a KT database, which is then used as a backend for several manual analysis (visual selection and classification steps) and automated analysis steps (channel alignment and filtering). From the database, all measures can be extracted. Here, we used localization counts per cluster and protein cluster distances to determine protein stoichiometry using a protein standard calibration and POI-cnp1[CENP-A] distances using Bayesian inference. **(B)** Using *E. coli* ferritin FtnA as a counting standard to calibrate POI copy numbers. Reconstructed SMLM image of isolated mEos3.2-A69T-FtnA oligomers. In our exemplary sample image, all assembly intermediates (monomers, dimers, 8mers) as well as final 24mers and some aggregates can be seen (exemplary 8mers, 24mers, and aggregates are highlighted with colored arrows). Scale bar, 500 nm. **(C)** Histograms of localization counts per selected 8mer (left) and 24mer (right) cluster. Using the mean (dashed lines) of 7.27 ± 2.72 for 8mers and 21.68 ± 10.28 for 24mers, we determined a calibration factor of 0.9. N = 1,458 (8mers) and N = 725 (24mers). **(D)** POI

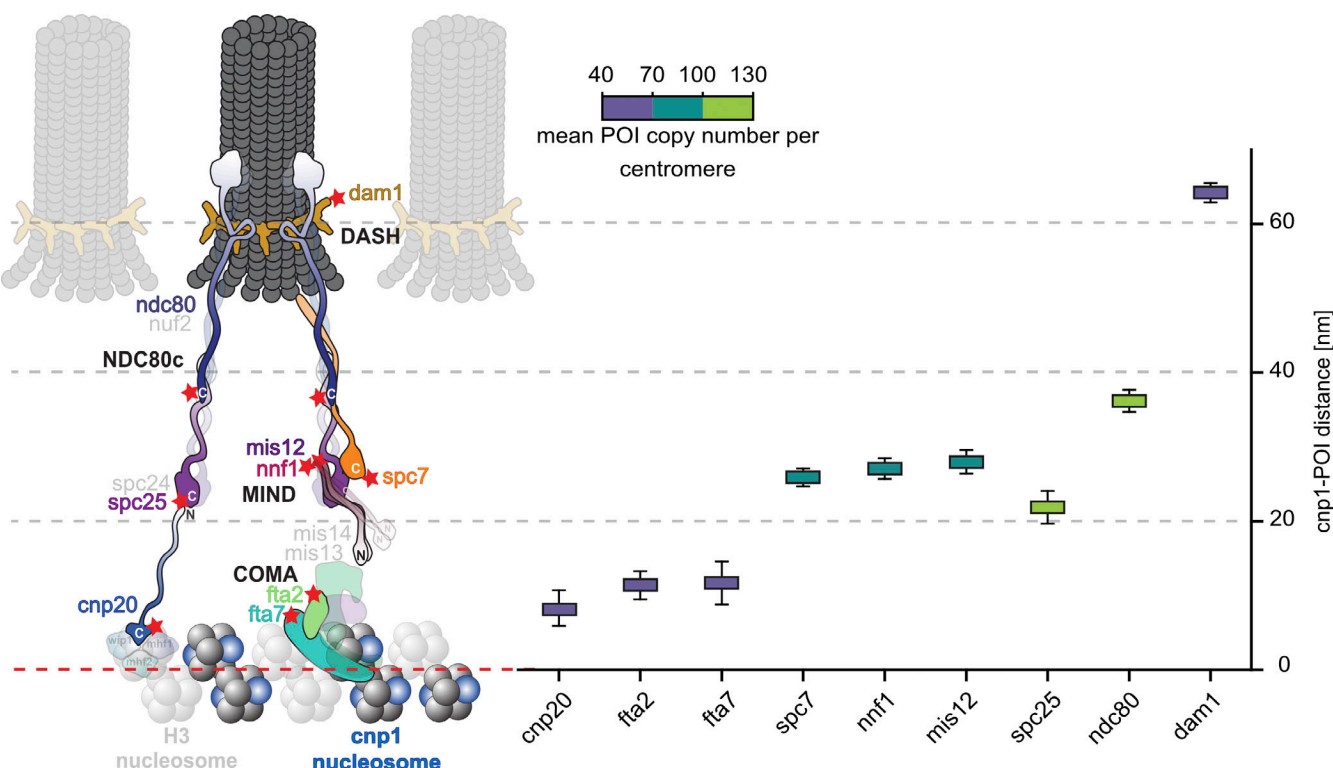

Figure 3. **POI-cnp1^CENP-A distances and protein stoichiometry within the KT complex.** Left: Schematic of the regional *S. pombe* centromere with parts of the inner and outer KT. Whenever information was available, the shapes of POIs and subcomplexes are shown according to cryo-EM or x-ray crystallography data. Red stars mark the position of the C-terminal fluorescent protein marker mEos3.2-A69T. Structures drawn at 25% opacity were not investigated. Right: POI distances to the reference cnp1^CENP-A and POI copy numbers per centromeric region as measured in this study. The colored boxes mark the mean and the whiskers the STD of the posterior probability density distribution for each cnp1^CENP-A-POI distance (Fig. 2 G, statistics in Table 1). The color of the box represents the mean POI copy number per cluster, as indicated by the scale bar (distributions of localization counts in Fig. 2 E, statistics in Table 2).

the MINDc binds spc7^KNL1 through mis14^NSL1 (Petrovic et al., 2010). This binding site, i.e., the C-termini of spc7^KNL1 and mis14^NSL1, is conserved (Petrovic et al., 2014). This places our in situ measured 1:1 ratio of nnf1^PMF1 and spc7^KNL1 in agreement with the in vitro literature and demands for future work on mis12.

Table 1. **Statistics of distance calculations**

| POI | Mean distance [nm] | STD [nm] | N |
|---|---|---|---|
| cnp20^CENP-T | 8.3 | 2.4 | 49 |
| fta2^CENP-P | 11.4 | 1.9 | 82 |
| fta7^CENP-Q | 11.7 | 2.9 | 58 |
| spc7^KNL1 | 25.9 | 1.2 | 215 |
| nnf1^PMF1 | 27.1 | 1.4 | 161 |
| mis12 | 28.0 | 1.6 | 102 |
| spc25 | 21.9 | 2.2 | 51 |
| ndc80^HEC1 | 36.2 | 1.5 | 135 |
| dam1 | 64.2 | 1.3 | 155 |

For each POI, the posterior mean distance, posterior STD, and the number N of KTs analyzed is listed. Full posterior distributions, approximated using Hamiltonian Monte Carlo (Materials and methods), can be found in Fig. 2.

Whereas the structure of the outer KT is conserved (D'Archivio and Wickstead, 2017; Meraldi et al., 2006; van Hooff et al., 2017), several strategies exist for bridging centromere and outer KT and the importance of the different inner KT structures, i.e., either based on cnp20^CENP-T, cnp3^CENP-C, or fta7^CENP-Q (COMAc), varies between organisms (Hamilton and Davis, 2020; van Hooff et al., 2017). While the inner KT of *Drosophila melanogaster* is defined by cnp3^CENP-C (Ye et al., 2016), *S. cerevisiae* relies on COMAc with cnp3^CENP-C as a backup (Hamilton et al., 2020; Hornung et al., 2014). Furthermore, while cnp20^CENP-T is non-essential in *S. cerevisiae* (Bock et al., 2012; De Wulf et al., 2003; Giaever et al., 2002; Schleiffer et al., 2012), it is the primary strategy in chicken (Hara et al., 2018) and humans (Suzuki et al., 2014), which also rely on cnp3^CENP-C as backup (Hamilton and Davis, 2020). To date, the preferred inner KT pathway of *S. pombe* is unknown (Hamilton and Davis, 2020). The deletion of cnp3^CENP-C in *S. pombe* is viable (Chik et al., 2019; Kim et al., 2010; Tanaka et al., 2009), whereas deletion of cnp20^CENP-T is not (Tanaka et al., 2009). This is inverted for *S. cerevisiae* (Bock et al., 2012; De Wulf et al., 2003; Giaever et al., 2002; Kim et al., 2010; Meeks-Wagner et al., 1986; Schleiffer et al., 2012). This organismal difference is also reflected in protein copy numbers as Mif2 (cnp3^CENP-C) is more abundant than Cnn1 (cnp20^CENP-T) in *S. cerevisiae*, chicken, and humans (Cieslinski et al., 2023; Johnston et al., 2010; Suzuki et al., 2015). For the COMAc,

**Table 2. Statistics of POI localization counts per centromeric region**

| POI | Mean count | STD | SEM | Median | Mean POI copy numbers* (±SE) | FP tag | N |
|---|---|---|---|---|---|---|---|
| cnp1$^{CENP-A}$ | 46.8 | 28.0 | 0.8 | 40.0 | ND | PamCherry1 | 1,245 |
| cnp20$^{CENP-T}$ | 55.0 | 32.5 | 4.3 | 47.0 | 61.1 ± 4.8 | mEos3.2-A69T | 57 |
| fta2$^{CENP-P}$ | 49.6 | 27.9 | 3.0 | 43.0 | 55.1 ± 3.3 | mEos3.2-A69T | 87 |
| fta7$^{CENP-Q}$ | 51.7 | 28.9 | 3.7 | 46.0 | 57.4 ± 4.1 | mEos3.2-A69T | 61 |
| spc7$^{KNL1}$ | 85.9 | 48.4 | 3.0 | 76.0 | 95.4 ± 3.3 | mEos3.2-A69T | 259 |
| nnf1$^{PMF1}$ | 83.3 | 47.2 | 3.2 | 70.0 | 92.6 ± 3.6 | mEos3.2-A69T | 217 |
| mis12 | 65.8 | 33.1 | 2.8 | 57.5 | 73.1 ± 3.1 | mEos3.2-A69T | 136 |
| spc25 | 107.2 | 52.2 | 5.6 | 49.5 | 119.1 ± 6.2 | mEos3.2-A69T | 86 |
| ndc80$^{HEC1}$ | 96.5 | 51.2 | 3.9 | 88.0 | 107.2 ± 4.3 | mEos3.2-A69T | 172 |
| dam1 | 57.3 | 33.1 | 2.2 | 49.5 | 63.7 ± 2.4 | mEos3.2-A69T | 232 |

Distributions can be found in Fig. 2 D and E. N, number of centromeric regions analyzed.
*Mean POI copy numbers were derived from the mean localization counts of each POI using 0.9 as a calibration factor as determined by the FtnA protein standard (Materials and methods).

fta7$^{CENP-Q}$ (Ame1) and mis17 (Okp1) are essential for both yeasts (Hayles et al., 2013; Kim et al., 2010) and deletions of fta2$^{CENP-P}$ and mal2$^{CENP-O}$ are non-viable in *S. pombe*.

In our measurements, cnp20$^{CENP-T}$ has a distance of 8.3 ± 2.4 nm to cnp1$^{CENP-A}$, similar to the MIND proteins fta7$^{CENP-Q}$ and fta2$^{CENP-P}$. Furthermore, we measured a 1:0.9 ratio of cnp20$^{CENP-T}$ to the COMAc (Table S4). This underlines the importance of cnp20$^{CENP-T}$ for *S. pombe* KTs as an essential protein (Tanaka et al., 2009) and suggests that *S. pombe* relies on the COMAc and cnp20$^{CENP-T}$ in equal measures—unlike *S. cerevisiae*, which relies on COMAc and cnp3$^{CENP-C}$.

To map the NDC80c, we measured ndc80$^{HEC1}$ and spc25, and obtained distances of 36.2 ± 1.5 nm and 21.9 ± 2.2 nm and copy numbers of 107.2 ± 4.3 and 119.1 ± 6.2, respectively. The shorter distance of spc25-C to cnp1$^{CENP-A}$ in comparison with the distances of the MIND proteins to cnp1$^{CENP-A}$ might reflect a spatial overlap of the POIs termini to interlock and stabilize the contact sites of the two sub-complexes and is in agreement with cross-linking data (Kudalkar et al., 2015). Furthermore, the order of increasing C-terminal distances from spc25 to spc7$^{KNL1}$, nnf1$^{PMF1}$, mis12, and ndc80$^{HEC1}$ correlates with a study that measured decreasing Fluorescence Resonance Energy Transfer signal between spc25 and the POIs (Aravamudhan et al., 2014). In another study, an increase of NDC80c and MINDc copy numbers from metaphase to anaphase was shown (Dhatchinamoorthy et al., 2017). For our data, we only selected and analyzed cells in metaphase/early anaphase. Comparing the mitotic spindle length and corresponding localization counts of POI and cnp1$^{CENP-A}$ clusters, we did not detect any interdependence of spindle length and copy numbers, which confirms our cell-cycle selection (data not shown).

For dam1 within the DAM1/DASHc, we found a distance of 64.2 ± 1.3 nm and a copy number of 63.7 ± 2.4. Comparing the positions of ndc80$^{HEC1}$ and dam1, dam1 is closer than one would expect when colocalizing with the N-term/globular head of ndc80$^{HEC1}$ (Aravamudhan et al., 2014; Roscioli et al., 2020). Dam1 possesses a highly variable length for different fungi and

whereas its C-terminus has been shown to interact with the globular head of ndc80$^{HEC1}$ in *S. cerevisiae*, it is predicted to be too short (155 aa in *S. pombe*, 343 aa in *S. cerevisiae*) to do so in *S. pombe* (Jenni and Harrison, 2018). Furthermore, deleting dam1 is viable for *S. pombe* (Zahedi et al., 2020) but inviable for *S. cerevisiae* (Giaever et al., 2002). Consistent with these observations, we hypothesize that *S. pombe* dam1 indeed interacts and colocalizes with the ndc80$^{HEC1}$-loop region. To explore this further, imaging the position of the N-terminus and loop region of ndc80$^{HEC1}$ or imaging dad1 (localizing at the DASH ring within the ndc80$^{HEC1}$ loop [Jenni and Harrison, 2018]) will be targeted in the future.

In contrast to the proteins of the outer KT complexes, which exhibit higher protein abundance, dam1 shows a similarly low copy number as the inner KT COMAc proteins. This was already seen in fluorescence measurements before (Joglekar et al., 2008), where protein abundance increased from the inner to the outer KT for both, *S. pombe* and *S. cerevisiae*, and a substantial drop was measured for *S. pombe* dam1 but not for *S. cerevisiae*.

## Summary

We used quantitative SMLM imaging to map the KT architecture at a nanometer resolution. It is the first holistic SMLM study of a multi-protein complex in *S. pombe* with which we were able to drastically improve the positioning and stoichiometry accuracy of 10 KT proteins and uniquely, within the full, native KT structure. Our work thus fills in the gap between the highly resolved in vitro studies and previous in vivo experiments and provides in situ nanometer-precise distances between the individual proteins and their copy numbers. Our measurements confirm the conserved structure of the outer KT and add knowledge on *S. pombe*–specific outer KT details, e.g., on dam1 localization at the DASH ring. For the inner KT, strategies vary in between different organisms, i.e., being based on either cnp20$^{CENP-T}$, cnp3$^{CENP-C}$, or fta7$^{CENP-Q}$. While other studies have reported the generally different use and importance of the inner KT structures between organisms, the *S. pombe* inner KT

architecture so far remained unexplored (Dimitrova et al., 2016; Hamilton and Davis, 2020; Petrovic et al., 2016; Przewloka et al., 2011; Screpanti et al., 2011). We quantified the stoichiometry of the inner KT pathways for *S. pombe* and showed the *S. pombe*–specific equal importance of COMAc and cnp20[CENP-T].

The SMLM study on the *S. cerevisiae* KT (Cieslinski et al., 2023), carried out in parallel to our work, is a perfect complement. Comparing the phylogenetically quite distant yeasts, the data consistently show that KTs generally possess similar architecture despite *S. cerevisiae* maintaining a point centromere and *S. pombe* a regional structure (Tables S4 and S5). Importantly, one substantial difference for the inner KT surfaced: the ratio of cnp20[CENP-T] to COMAc is 1:0.9 in *S. pombe* and 1:2.0 in *S. cerevisiae*. This is a strong indication that cnp20[CENP-T] organization indeed differs between the two organisms. Furthermore, the positioning of *S. cerevisiae* Ask1 is consistent with the position of *S. pombe* dam1 and thus supports our reasoning that the C-terminus of *S. pombe* dam1 is localized at the DASH ring and not at the ndc80 globular heads (like for *S. cerevisiae*).

In the outlook, our strain library and results provide an excellent platform for quantitative SMLM work on the KT that goes beyond copy numbers and distances, and we will—by further dissecting our *S. pombe* KT data—focus on structural shapes and changes during the cell cycle, as well as extend our work to medically relevant mutants.

## Materials and methods
### Choosing a labeling strategy for quantitative SMLM in *S. pombe*
While the number of super-resolution microscopy studies in fission yeast being published has been increasing in recent years, the majority of them are single color studies that target individual proteins (Akamatsu et al., 2017; Bell et al., 2014; Etheridge et al., 2014; Etheridge et al., 2021; Lando et al., 2012; Laplante et al., 2016; Matsuda et al., 2015), and only a few are dual-color studies visualizing several targets at the same time (Bestul et al., 2021; McDonald et al., 2017; Virant et al., 2017). Among those studies, SMLM imaging takes on a special role, as this technique does not only offer increased resolution but also allows the determination of protein stoichiometries (Akamatsu et al., 2017; Lando et al., 2012; Laplante et al., 2016). Due to the limited number of working fluorophore/labeling combinations for SMLM imaging in microbial organisms, developing a successful multicolor single-molecule localization microscopy strategy can be challenging (Vojnovic et al., 2019).

To construct a nanoscale map of the KT in *S. pombe*, we labeled three proteins, the protein of interest, the centromeric reference cnp1[CENP-A], and the spindle pole protein sad1. Imaging the spindle poles enabled us to (a) easily identify and read out the correct cell-cycle stage and focal plane and (b) drastically improve the accuracy of our distance calculations. The latter is possible as all three protein(-clusters) can be assumed to lie on a straight line, which allows a direct estimation of the size of the measurement errors from the deviation of that line (see data analysis and Data S1).

As all organisms and targets have different requirements and specifics, SMLM labeling strategies, which need to be highly optimized to produce satisfying results, must be specifically adapted and tested for the actualities of each biological system (Vojnovic et al., 2019). In *S. pombe*, we avoided the green and blue color channels, due to increased phototoxicity as well as the fact that we found high levels of autofluorescence in those channels, as *S. pombe* is easily stressed by unstable or insufficient pH, temperature, or aeration.

Furthermore, we found a high level of autofluorescence due to a selection marker in standard laboratory *S. pombe* strains, which works by interrupting the adenine biosynthesis pathway to accumulate a precursor molecule—a bright red pigment which can be easily seen in screening colonies (Allshire, 1995; Levenberg and Buchanan, 1957; Lukens and Buchanan, 1959). These adenine-deficient laboratory strains (most common are ade6-M210 and ade6-M216), even when grown under full adenine supply, possess a background level of metabolites which strongly disturb sensitive SMLM measurements even when colonies or liquid cultures remain inconspicuously colorless (Fig. S1 and Winkelmeier, 2018). We thus "cured" the adenine gene using a WT strain template for all strains in this study.

For three labels, we were in need for three fluorophores with mutually supporting SMLM imaging properties: In general, in situ targeting of proteins can be achieved extrinsically by using dyes (e.g., using immunofluorescence and enzyme or peptide tags) or intrinsically by using FPs as genetic tags. On the one hand, organic dyes are brighter and therefore have a higher signal-to-noise ratio and localization precision, but on the other hand they bring two disadvantages: (a) extrinsic staining is accompanied by non-specific staining and has unknown and possibly even heterogeneous staining efficiency, and (b) SMLM imaging of organic dyes requires imaging buffers and might exhibit heterogeneous switching behavior. These drawbacks significantly affect any quantitative SMLM study but become especially severe for low copy number targets and for multi-color studies. While evaluating different labeling and imaging strategies, we tested different dyes. Surprisingly, the "golden SMLM standard" Alexa Fluor 647 does not work reliably in *S. pombe* (highly heterogeneous blinking to even non-blinking behavior—which we not only found for live cells but also for fixed cells and for all buffers tested; see buffer table in Turkowyd et al. [2016]) for reasons unknown to us and exhibits extraordinary high nonspecific staining levels under even rigorous staining protocols due to its unmasked charges (Fig. S1). Other dyes yielded the desired low nonspecific staining and controlled blinking behavior. Nevertheless, dyes with masked charges (e.g., CF680) never achieved high labeling efficiencies, which may be partially attributed to their increased size (e.g., CF680 has about 2.3 times the molecular weight of Alexa Fluor 680 due to the masking groups) and the high molecular crowding of *S. pombe* cells (Fig. S1). Among all tested dyes, the best working ones available to us were JF549 and CF647 (Fig. S1). Nevertheless, for all labeling strategies involving dyes, achieving reliable and high labeling efficiencies as well as quantitative dual-color SMLM readouts without any channel crosstalk or a high miss rate of events (we tested chromatic dual-color,

spectral demixing, and dye-FP-mixture strategies) remained problematic. Thus, all measurement strategies in our final selection solely relied on direct FP fusions and were all in the orange-red part of the visual spectrum. As mEos2, our first FP of choice induced conspicuous phenotypical anomalies for several POIs (and is known to artificially aggregate and self-oligomerize at higher densities interfering with POI positioning and function [Wang et al., 2014]), we carefully tested different FPs and ultimately decided to utilize a dual-color approach published in Virant et al. (2017), which relies on mEos3.2-A69T and PAmcherry1 and added mScarlet-I as the label for the SPB reference.

### Strain construction

To construct a dual-color reference strain with both a reference at the centromere and at the SPB, we added a spindle reference to the *S. pombe* strain DL70, which already carries a PAmCherry1-cnp1[CENP-A] N-terminal fusion (Lando et al., 2012). To tag the SPB protein sad1 with mScarlet-I (Bindels et al., 2017) at its C-terminus, we adapted a cloning strategy from Hayashi et al. (2009): DL70 cells were made chemically competent (Frozen-EZ Yeast Transformation II Kit, #T2001; Zymo Research) and were transformed with a linear DNA fragment consisting of the mScarlet-I sequence and a hygromycin resistance cassette flanked by 500–600 bp of homology arms on both sides for homologous recombination into the genome downstream of the native sad1 gene. Additionally, we inserted the terminator ADH1 from *S. cerevisiae* (Curran et al., 2013) between POI-FP and resistance genes to ensure native protein expression independent from the resistance gene. Cells were streaked on a Yeast Extract with Supplements (YES) agar plate, incubated overnight at 32°C, and replica-plated on a YES agar plate containing 100 mg/liter hygromycin B (#10843555001; Roche). The hygromycin plates were incubated for 1–2 d at 32°C until colonies had formed. Colonies were checked for successful integration of the mScarlet-I ADH1 hygromycin cassette by colony PCR and subsequent sequencing. When we tested the resulting strain SP129 for single-molecule imaging, we noted a high level of autofluorescence that we could attribute to an accumulating precursor in the interrupted adenine biosynthesis pathway due to the ade6-M210 mutation, which is as bright as the single-molecule FP intensities and therefore severely compromises the SMLM readout (Fig. S1 and Materials and methods; Allshire, 1995; Levenberg and Buchanan, 1957; Lukens and Buchanan, 1959). We thus repaired the ade6 gene using a WT strain template. Transformed cells were streaked on Edinburgh Minimal Medium (EMM) 5S agar plates and grown overnight at 32°C, replica-plated on EMM agar plates containing 225 mg/liter of histidine, leucine, lysine, and uracil each, but only 10 mg/liter of adenine and incubated for 1–2 d at 32°C. Cells that still possessed the truncated ade6 gene turned pink due to accumulation of the precursor (which is also used as a selection marker in the literature [Allshire, 1995]). Colonies with a successfully integrated WT ade6 gene stayed white and were checked by sequencing. This new strain (SP145) was used as the parent strain for the construction of a three-color KT library: Here, each POI was tagged with the FP mEos3.2-A69T using the same procedure as for sad1-tagging but using a kanamycin resistance cassette (Fig.

S2 A). Successful transformants were selected on YES agar plates containing 100 mg/liter geneticin (#4727894001; Roche) and checked with colony PCR and sequencing. Sequencing was performed with the Mix2Seq kit (Eurofins Genomics). A colony was picked for each newly generated strain, cultured to early stationary phase in rich YES medium, mixed in a 1:1 ratio with sterile 87% glycerol, and stored at –80°C. All strains used in this study can be found in Table S1, and all primers used for strain construction can be found in Table S2.

### Spot tests

To assess the effect of the FP tags on the POIs and reference proteins, spot tests were conducted to compare the temperature and TBZ (inducing mitotic defects by microtubule depolymerization) sensitivity to the parental h+ WT strain (Fig. S2). For this, *S. pombe* strains from cryostocks were streaked on YES agar plates and incubated at 32°C for 2 d. After growing the overnight cultures in YES medium at 25°C to an $OD_{600}$ of 1–2, all cultures were adjusted to an $OD_{600}$ of 1.0 and 2 µl each of a serial dilution of 1:1, 1:10, 1:100, and 1:1,000 were plated either on YES agar plates and incubated at 25, 32, or 37°C for 3 d for temperature sensitivity testing, or on YES agar plates containing 2.5, 5.0, 7.5, or 10.0 µg/ml TBZ (#23391; Cayman Chemical Company) and incubated at 25°C for 3 d.

### Flow cytometry

To assess the effect of the fluorescent protein tags on the phenotype of cells, flow cytometry was conducted (Fig. S2). Cells were streaked from cryostocks on YES agar plates and grown at 32°C for 2–3 d until colonies were visible. A single colony was picked, inoculated in 10-ml YES medium and grown overnight (25°C, 180 rpm). Subsequently, the $OD_{600}$ for each strain was measured and an appropriate cell amount was used to inoculate 10 ml of EMM 5S to the same starting $OD_{600}$. Cells were grown overnight (25°C, 180 rpm) and the $OD_{600}$ was measured again the next morning to verify that cells were in logarithmic growth phase. Light scattering (forward scatter [FSC] and side scatter [SSC]) was measured with a BD LSRFortessa flow cytometer (BD Biosciences). Before each measurement, samples were mixed thoroughly to disperse cell aggregates. For each strain, 10,000 cells were recorded using the FACSDiva software (BD Biosciences). Data were analyzed and plotted with the software FlowJo (BD Biosciences).

### Sample preparation for SMLM imaging

After streaking *S. pombe* strains from cryostocks on YES agar plates and incubating them at 32°C until colonies were visible, overnight cultures were inoculated in rich YES media at 25°C and grown to an $OD_{600}$ of 1–2. Next, 2× 50 ml night cultures were inoculated in freshly prepared EMM 5S media starting at $OD_{600}$ 0.1 and grown at 25°C for around 16 h to $OD_{600}$ 1.0–1.5. Cultures were centrifuged at 2,000 rpm at 25°C for 5 min, the supernatant was discarded, and the pellets were resuspended in 200 µl fresh EMM 5S each. Meanwhile, for cell synchronization, two previously prepared and frozen 15 ml falcons containing 20% lactose (#6868.1; Carl Roth) were thawed at 30°C, and the cell suspension was carefully added on top of the still cold lactose

gradient (Fig. 2). The falcons were then centrifuged at 1,000 rpm for 8 min and 1 ml of the resulting upper cell layer, which consists of cells in early G2 stage, was extracted, added to 1 ml fresh EMM 5S and centrifuged at 4,000 × g for 1 min. The pellets were resuspended in 500 µl EMM 5S, combined and the $OD_{600}$ was measured. The cell suspension was inoculated in fresh EMM 5S at a starting $OD_{600}$ of 0.6 and grown for 60, 70, or 80 min at 25°C. At each time point, 2 ml aliquots were taken from the main culture for chemical fixation with 3% PFA (#F8775; Sigma-Aldrich) at 20 min at RT. The fixed cells were centrifuged at 4,000 × g for 2 min and washed six times with 1 ml washing buffer alternating between 1xPBS, pH 7.5 (#70011-036; Gibco), and 50 mM Tris-HCl, pH 8.5 (#AE15.2; Carl Roth), with each 10 min of incubation. The cells were resuspended in 500 µl Tris-HCl after the last wash, a 1:1,000 dilution of previously sonicated dark red (660/680) FluoSpheres (#F8807; Molecular probes) was added to serve as fiducial markers for drift correction (Balinovic et al., 2019) and incubated for 10 min in the dark. Ibidi 8-well glass bottom slides (#80827; Ibidi) were cleaned by incubating 1M KOH for 30 min, coated with poly-L-lysine (#P8920-100ML; Sigma-Aldrich) for 20 min, and air-dried thereafter. Next, 200 µl of the bead/cell mix were added to each well and were incubated for 10 min in the dark. Meanwhile, a 0.5% wt/vol low gelling agarose solution (#A9414-100G; Sigma-Aldrich) was melted in 50 mM Tris-HCl at 70°C for 1–2 min, a 1:1,000 dilution of dark red (660/680) FluoSpheres was added, and the mix was kept at 32°C until further use. The supernatant of the bead/cell mix incubating in the Ibidi 8-well glass bottom slide was carefully removed on ice, and six drops of the agarose mix were carefully added on each well and incubated for 5 min for proper gelation. Finally, 200 µl 50 mM Tris-HCl, pH 8.5, was added to each well. Samples were stored at 4°C for a maximum of 1 d until being imaged.

### Microscope setup and SMLM Imaging

A custom-built setup based on an automated Nikon Ti Eclipse microscopy body including suitable dichroic and emission filters for SMLM imaging using Photoactivated Localization Microscopy imaging by primed photoconversion (PC-PALM; ET dapi/Fitc/cy3 dichroic, ZT405/488/561rpc rejection filter, ET610/75 bandpass, all AHF Analysentechnik) and a CFI Apo TIRF 100× oil objective (NA 1.49; Nikon) was used for multi-color imaging experiments as described earlier (Virant et al., 2017). The Perfect Focus System (Nikon) was utilized for z-axis control, and all lasers (405 nm OBIS, 488 nm Sapphire LP, 561 nm OBIS, Coherent, Inc.) except the 730 nm laser (OBIS, Coherent, Inc.) were modulated by an acousto-optical tunable filter (AOTF; TF525-250-6-3-GH18, Gooch and Housego) in combination with the ESio AOTF controller (ESImaging). Single-molecule fluorescence was detected using an emCCD camera (iXON Ultra 888, Andor) at a pixel size of 129 nm, and the image acquisition was controlled by a customized version of µManager (Edelstein et al., 2010). All imaging was performed at RT.

Movie acquisition of all 3C strains was performed sequentially in HILO mode (Tokunaga et al., 2008). First, the sample was illuminated at low intensity 561 nm with 0.1–1 W*cm⁻² at the sample level to identify fields of view containing a good number of cells with two SPB spots (metaphase and early anaphase A cells) aligned in parallel to the focal plane. The sad1-mScarlet-I signal was read-out by taking 100 images of 60 ms exposure time with the 561 nm laser at 200 W*cm⁻² (at the sample level). Residual mScarlet-I signal was photobleached using 1 kW*cm⁻² of 561 nm laser light for 30–60 s. Next, the green-to-red photoconvertible FP mEos3.2-A69T was imaged in the same spectral channel using primed photoconversion by illuminating the sample with 488 (at every 10th frame) and continuous 730 nm laser light (6–70 & 450 W*cm⁻², respectively) and readout by 561 nm at 1 kW*cm⁻² for 15,000 frames of 60 ms exposure. An additional post-bleaching step with 488 nm laser light (210 W*cm⁻² for 20 s) was performed to ensure that no residual mEos3.2-A69T signal was detected in the next acquisition step. Finally, PAmCherry1-cnp1$^{CENP-A}$ was photoactivated by illumination with 10–20 W*cm⁻² of 405 nm laser light and readout by 561 nm at 1 kW*cm⁻² for 10,000 frames of 60 ms exposure. Keeping the field of view well defined, we ensured homogeneous illumination.

### Staining of samples with organic dyes

The construction of the N-terminal Halo-cnp1 strain is described in Vojnovic (2016). An overnight Halo-cnp1 YES culture was harvested by centrifugation, resuspended in fresh YES, and fixed with 3.7% PFA at RT for 10 min. Afterwards, the sample was washed 3× and residual PFA was quenched using 1× PEM (100 mM Pipes [#P1851-100G; Sigma-Aldrich], pH 6.9, 1 mM EGTA [#EDS-100G; Sigma-Aldrich], and 1 mM $MgSO_4$ [#M2643-500G; Sigma-Aldrich]) containing 50 mg/ml $NH_4Cl$ (#P726.1; Carl Roth). Next, the sample was permeabilized using 1.25 mg/ml Zymolyase (#320921; MP Biomedicals) in 1× PEM at 37°C for 10 min and subsequently washed 3× with 1× PEM for 10 min. The sample was incubated using a few drops of Image-iT FX signal enhancer (#I36933; Invitrogen) at RT for 1 h to prevent nonspecific staining and afterwards stained with 50 nM Halo-CF647 in PEMBAL buffer (1× PEM containing 3% BSA [#A8549-10MG; Sigma-Aldrich], 0.1% NaN₃ [#4221.1; Carl Roth], and 100 mM lysine hydrochloride [#L5626; Sigma-Aldrich]) at RT for 3 h. The stained cells were then washed four times for 10 min with alternating 1× PEM and 1× PBS and incubated at RT for 15 min on a previously washed and with poly-L-lysine–coated Ibidi 8-well glass bottom slide. Finally, the attached cells were washed twice with 1× PEM for 10 min.

### SMLM imaging and post-processing of dye-stained samples

The stained Halo-cnp1 strains were imaged in a redox buffer (100 mM MEA [#MG600-25G; Sigma-Aldrich] with an oxygen scavenger system [van de Linde et al., 2011] in 1× PEM) and illuminated with a 637 nm laser (OBIS, Coherent, Inc.) at 1.4 kW*cm⁻² at the sample level with an exposure time of 50 ms for 10,000 frames on the same setup as described before, except that a different dichroic (Chroma; ZET 405/488/561/640) and longpass filter (AHF; 655 LP) were used. Single-molecule localizations were obtained by the open-source software Rapidstorm 3.2 (Wolter et al., 2012). In order to track localizations that appeared in several frames and thus belong to the same emitting molecule, the NeNA localization precision value (Endesfelder

et al., 2014) was calculated in the open-source software Lama (Malkusch and Heilemann, 2016).

### Reconstruction of SMLM images

For visualization, we aimed to reconstruct SMLM images that neither over- nor under-interpret the resolution of the SMLM data and resemble fluorescence images as closely as possible. Localizations were tracked together using the Kalman tracking filter in Rapidstorm 3.2 with two sigma, and the NeNA value used as sigma (dSTORM data), or using kd-tree tracking (3C strain library data, described in detail in the Data analysis section below). Images were then reconstructed in Rapid-storm 3.2 or 3.3 with a pixel size of 10 nm. Rapidstorm linearly interpolates the localizations and fills the neighboring pixels based on the distance between the localization and the center of the main subpixel bin to avoid discretization errors (Wolter et al., 2012). These images were then processed with a Gaussian blur filter based on their NeNA localization uncertainty in the open-source software ImageJ 1.52p (Schindelin et al., 2012). Importantly, the images were only used for image representation purposes, and all data analysis steps were conducted on the localization data directly (refer data analysis).

### Construction of the FtnA protein standard

The pRSETa-FtnA backbone was amplified from pRSETa BC2-Ypet-FtnA (Virant et al., 2018) and the mEos3.2-A69T fragment was amplified from pRSETa-mEos3.2-A69T (Turkowyd et al., 2017) with an 18 and 23 bp overlap to the pRSETa-FtnA backbone, respectively. Purified DNA fragments (Clean & Concentrator-5 Kit, #D4013; Zymo Research) were merged by Overlap-Extension PCR in a 1:2 ratio of backbone to insert. The resulting plasmid was purified and transformed into competent DH5α cells and incubated on LB AmpR agar plates at 37°C over-night. Plasmids were isolated from individual colonies (Monarch Plasmid Miniprep Kit, #T1010L; New England Biolabs) and sequenced (Mix2Seq Kit NightXpress, Eurofins genomics). The confirmed construct was transformed into competent Rosetta DE3 cells.

### Sample preparation and imaging of the FtnA protein standard

Rosetta DE3 containing pRSETa mEos3.2-A69T-FtnA were grown over night at 37°C and 200 rpm in 50 ml AIM (auto-inducing medium containing freshly added 25 g/liter glucose [#X997.2; Carl Roth] and 100 g/liter lactose [#6868.1; Carl Roth]). Cells were harvested by centrifugation at 4,000 × $g$ and 4°C for 30 min, and the pellet was resuspended in 2 ml 50 mM Tris HCl (#AE15.2; Carl Roth) containing 25 mM NaCl (#S3014-500G; Sigma-Aldrich) at pH 8.5. The bacterial cell wall was digested by incubation with 50 mg/ml Lysozyme (#L2879; Sigma-Aldrich) for 2 h at 4°C. The sample was homogenized (Tough micro-organism lysing tubes [#VK05; Bertin] with 0.5-mm glass beads in Precellys 24 tissue homogenizer at 3 × 15 s and 5,000 rpm at RT). The cell lysate was centrifuged at 4,000 × $g$ for 30 min at 4°C, and the supernatant was transferred into a clean Eppendorf tube. The centrifugation and transfer were repeated to ensure a debris-free supernatant. For long-term storage, 30%

glycerol (#G5516; Sigma-Aldrich) was added and the stock was kept at −20°C until further use.

In order to prepare the FP-FtnA oligomer surfaces, the mEos3.2-A69T-FtnA stock was diluted 1:5,000 in 50 mM Tris HCl containing 25 mM NaCl at pH 8.5 and incubated with a sonicated 1:1,000 dilution of dark red (660/680) FluoSpheres (#F8807; Molecular probes) on a clean, poly-L-lysine coated Ibidi 8-well glass bottom slide for 10 min. The sample was washed twice using 50 mM Tris HCl containing 25 mM NaCl at pH 8.5 and 0.5% wt/vol low gelling agarose solution (#A9414-100G; Sigma-Aldrich) was carefully added to the top to ensure proper immobilization. After gelation for 5 min on ice, 300 μl of 50 mM Tris HCl containing 25 mM NaCl at pH 8.5 were added to the imaging well.

The same microscope setup and imaging routine that was used for imaging the 3C strain library was utilized to image the isolated mEos3.2-A69T-FtnA oligomers. To simulate the mScarlet-I imaging and bleaching step prior to mEos3.2-A69T imaging, the sample was first illuminated with the 561 nm laser at 1 kW*cm$^{-2}$ for 1 min before the PC-PALM read-out of mEos3.2-A69T was recorded.

### Data analysis

An overview of the workflow can be found in Fig. 2.

#### Localization, drift correction, and filtering

First, all movies were localized using ThunderSTORM (Ovesný et al., 2014) with the B-spline wavelet filter with a scale of 2.0 pixel and order of 3, an 8 pixel neighborhood and threshold of 1.5*std(wavelet), the integrated Gaussian point spread function (PSF) model with a sigma of 1.4 pixel and a weighted least squares optimization with a fit radius of 4 pixel. The fluorescent beads embedded in the sample were used for drift correction. The combined drift trace as well as the drift trace for each bead were saved and checked for inconsistencies. In those instances, the localization file was manually drift corrected using a custom script written in Python 3 that allowed the selection of individual beads.

A nearest neighbor tracking algorithm based on kd-tree (Jones et al., 2001) was run on the localization files. For each localization, the nearest neighboring localization in a 150 nm radius was identified in the first subsequent frame. Neighbor identifiers were stored and used to connect localizations into tracks. Tracks were then merged by averaging the coordinates, intensity, and PSF width of all localizations in a track. Averaged localizations were filtered based on these parameters to discard statistical outliers (Q1 – 1.5*IQR, Q3 + 1.5*IQR) in intensity and chi-square goodness of fit. For the PSF width, a set threshold of a sigma of 70–200 nm was used.

#### Visualization and manual analytics

For several manual steps, localizations were visualized in a custom software able to flexibly zoom in/out and to switch be-tween/overlay sad1/POI/cnp1$^{CENP-A}$ channels. Using this tool, individual localizations could be selected and classified. For channel alignment, localizations belonging to the same fiducial marker in all three channels were grouped together. Cells with

visible KT protein clusters in the focal plane were selected and classified as individual region of interests, and all clusters were annotated. cnp1$^{CENP-A}$ clusters were paired together with corresponding POI clusters. Whenever there was any doubt whether two clusters belonged to the same KT or whether a cluster represented a single centromere region or several, the clusters were discarded. Two exemplary datasets can be found in Data S2. The annotation work was quality-checked by cross-checking the annotation of two different persons.

### Channel alignment

The alignment between two channels along a given axis was calculated as the inverse-variance weighted mean displacement of beads present in both channels. The position of a bead in each channel was calculated as the mean position of all localizations associated with the given bead in this channel. The variance of the displacement is then given by the sum of the squared standard errors of the two positions. The alignment of two channels was deemed viable if it was based on at least two beads and a minimum of 15,000 (channels 1 and 2) or 180 (channel 3) localizations. Initially, only beads with $e^{-|\log(FWHM_x/FWHM_y)|} > 0.9$ were used.

For all movies where this did not result in a viable alignment, this threshold was successively lowered to 0.8, 0.7, 0.6, and 0.5 until a viable alignment was found. Movies for which no viable assignment could be found using this procedure were not taken into consideration when estimating the cnp1$^{CENP-A}$-POI distances.

### Distance calculation

We used Bayesian inference to determine the real distance between cnp1$^{CENP-A}$ and the POI from the measured centers of their respective clusters, assuming Gaussian measurement errors of uncertain size. To improve accuracy, we also took the position of the associated spindle pole into account, as the three points (centers of POI, cnp1$^{CENP-A}$, corresponding sad1 cluster) can be assumed to lie on a straight line. As it is not possible to reliably determine which spindle pole a centromere is attached to, we built a mixture model to take both possibilities into account. (The SPB closest to a KT is not necessarily the SPB to which the KT is attached to. The reasons for this are that *S. pombe* undergoes closed mitosis where the nuclear envelope does not break up before chromosomes separate and, furthermore, KTs oscillate back and forth along the mitotic spindle until DNA segregation takes place [Rieder et al., 1986]). To formalize the idea that a centromere is more likely to be attached to the closer spindle pole, we calculated a mixture coefficient $\lambda = 1/(1 + (d_2/d_1)^{-3})$, where $d_1$ and $d_2$ are the measured distances between cnp1$^{CENP-A}$ and the first and second spindle poles, respectively. The coefficient then corresponds to the prior probability that the centromere is attached to the first spindle pole. We visualize the model in Fig. 2.

We considered three types of Gaussian measurement errors: the measurement error of the spindle, the measurement error of the center of the POI and cnp1$^{CENP-A}$ clusters, and the error of the alignment of the different channels. As the localizations belonging to a single spindle pole are (approximately) normally distributed, the variance of their mean can be estimated with high precision from the sample variance in *x* and *y*. The

variances of the errors of the clusters and the alignment are more difficult to estimate directly and were therefore estimated together with the cnp1$^{CENP-A}$-POI distances, using a scaled inverse $\chi^2$ distribution as prior. Whereas the measurement error can be different for each individual cluster, the alignment error was assumed to be identical for all clusters in a movie. For the alignment error, we estimated the scale parameter $\tau^2$ of the prior distribution from the position of the beads in different channels (although this is likely to be an underestimation, as the same beads were used to calculate the alignment) and the degrees of freedom $\nu$ from the number of beads used in the movie. We used constant values $\tau^2 = 100$ nm$^2$ and $\nu = 4$ for the error of the cluster centers, as they seemed to work reasonably well upon visual inspection. A more objective estimation of this error is difficult, as it results mainly from errors in the clustering, which was performed manually.

We used the Hamiltonian Monte Carlo algorithm as implemented by Stan (Carpenter et al., 2017; Stan Development Team, 2021) and the CmdStanPy python package (Stan Development Team, 2019) to approximate the posterior distribution of the cnp1$^{CENP-A}$-POI distances according to our model (code in Data S1 and on GitHub: https://github.com/Endesfelder-Lab/Kinetochore_Distances). Posterior distributions are shown in Fig. 2, and means are plotted in Fig. 3 and summarized in Table 1.

### Protein stoichiometry

SMLM localization counts do not directly reflect on the real POI copy numbers due to several over- and undercounting factors (e.g., incomplete FP maturation or photoconversion or FP blinking [Turkowyd et al., 2016]). A robust way to acquire statistically correct average POI copy numbers is to calibrate the localization count data with corresponding data of a protein standard of known stoichiometry using the same sample, imaging, and analysis protocols as for imaging the POI. We utilized the *Escherichia coli* protein ferritin (FtnA), a homo-oligomeric protein standard of 24 subunits labeled by mEos3.2-A69T to calibrate our POI localization counts (Finan et al., 2015; Virant et al., 2018). FtnA assembles from monomers into dimers to form 8mers which then arrange into the final 24mer structure. As depicted in Fig. 2, FtnA assembly intermediates (FtnA monomers, dimers, 8mers), full 24mers as well as aggregates can be identified by eye and their intrinsic stoichiometric differences are sufficient to discriminate them by fluorescence intensities. We manually selected 1,458 8mers and 725 24mers from 29 imaging regions of interest from two imaging days. Their localization count distribution yields a mean of 7.27 ± 2.72 standard deviation (STD) and 21.68 ± 10.2 STD counts per 8mer and 24mer cluster, respectively (Fig. 2). Thus, both give a correction factor of about 0.9 to translate localization counts into POI counts. With the help of this factor, all POI counts were translated into POI copy numbers per centromeric cluster as given in Table 2.

### Verifying the reliability of measured localizations counts and distances

In all measurements, we quantitatively assessed not only the localization counts for the POI clusters but also the counts of the

corresponding PAmCherry1-cnp1$^{CENP-A}$ cluster. We use these reference counts to identify possible inconsistencies in the sample preparations and imaging routines of the different experiments. The PAmCherry1 localization counts for the cnp1$^{CENP-A}$ reference remained constant independent of individual sample preparations, imaging days, and different strains as can be seen in Fig. 2, where the data are exemplarily sorted into the different strain categories. The POI protein copy numbers as given in Table 2 show a large coefficient of variation. To assess to which extend this variability reflects a technical inability to measure protein levels accurately or some flexibility in KT protein stoichiometry (e.g., due to differing numbers of kMTs per KT), we can use the data of the FtnA oligomer counting standard: The FtnA oligomer is a biologically highly defined structure. Thus, our FtnA measurements can directly serve as a proxy for the contribution of the technical inaccuracy of our PALM imaging and analysis strategy to the variance. Using the results of 21.68 counts ± 10.2 STD for the 24mers and of 7.27 counts ± 2.72 STD for the 8mers, we can estimate that the technical inaccuracy causes a coefficient of variance of 0.35–0.5, thus almost completely explaining the experimentally seen coefficient of ~0.5 for our POI data (Table 2). Due to this high technical inaccuracy, we cannot resolve sub-populations of possibly different KT structures (and thus POI copy numbers) on 2–4 kMTs in our current counting data (Fig. S3).

For distances, we examined whether the POI-cnp1$^{CENP-A}$ distance depends on the mitotic spindle length, e.g., due to different phases in chromosome separation or different forces. As seen in Fig. S3 for the POI dam1 (but being true for all our measurements [data not shown]), we do not detect any dependence of the centroid distance to the mitotic spindle length and therefore conclude that the KTs, as we analyzed them, all show the same layers and stretching. This is further validated by the posterior distributions of POI-cnp1$^{CENP-A}$ distances in Fig. 2, which consistently show only a single, well-defined peak. Furthermore, using a nup132-GFP strain in early G2 phase, we measured the average nuclear diameter of *S. pombe* to be 2.4 ± 0.19 μm (data not shown), in agreement with the literature (Maclean, 1964; Toda et al., 1981). The spindles included in our analysis have spindle lengths well below the nuclear diameter, thus excluding anaphase B cells (Fig. S3). Of additional note: While for *S. pombe* we assume—based on literature data and atb2 MT imaging (data not shown)—that cnp1, POI, and one spindle indeed lie on a line in metaphase/early anaphase, this has to be revisited for higher eukaryotes, where the outer KT domain has been observed to swivel around the inner KT/centromere, an effect thought to facilitate kMT attachment and which only reduces in anaphase (Smith et al., 2016).

### Online supplemental material

Fig. S1 shows autofluorescence of metabolites and dye staining in *S. pombe*, related to Materials and methods. Fig. S2 shows the cloning strategy, sample preparation, and strain health assessment used in this study, related to Materials and methods. Fig. S3 shows the independence of the centroid distances from the mitotic spindle length and the angular offset of kMTs and spindle axes, related to Materials and methods. Table S1 shows a list of *S. pombe* and *E. coli* strains used in this study, related to Materials and methods. Table S2 shows a list of oligonucleotides used in this study, related to Materials and methods. Table S3 shows the weighted mean of localization counts for POIs belonging to the same KT subcomplex. Table S4 shows a comparison of POI copy number ratios between different KT subcomplexes with the literature. Table S5 shows a comparison of protein cluster distances of this study and Cieslinski et al. (2023). Data S1 contains the STAN code and raw distance data for distance analysis. Data S2 contains two exemplary annotated kinetochore data sets and a short info.txt.

### Data availability

The Stan code and our raw data on the posterior distributions of KT protein distances can be found in Data S1.

## Acknowledgments

We are grateful to E. Laue for discussions and advice, to M. Rigl, K. Schmidt, and B. Aybey for their help in the laboratory, and to S. González Sierra and V. Sourjik for their help with flow cytometry.

This work was supported by funds from the Max Planck Society, a travel grant to D. Virant by the Boehringer Ingelheim Fonds for a research stay with D. Lando, funds at Carnegie Mellon University, the National Science Foundation AI Institute: Physics of the Future, National Science Foundation PHY-2020295 and funds at Bonn University. Open access funding provided by the Max Planck Society.

Author contributions: D. Lando and U. Endesfelder designed research; D. Virant, I. Vojnovic, J. Winkelmeier, D. Lando, and U. Endesfelder designed experiments; D. Virant, I. Vojnovic, and J. Winkelmeier performed experiments; D. Virant, B. Turkowyd, M. Endesfelder, and U. Endesfelder developed analysis procedure and software; I. Vojnovic, J. Winkelmeier, M. Endesfelder, and U. Endesfelder performed the all-data final analysis; D. Virant and U. Endesfelder acquired funding; I. Vojnovic and J. Winkelmeier designed figures and tables; U. Endesfelder wrote the paper with input from all authors.

Disclosures: The authors declare no competing interests exist.

Submitted: 24 September 2022

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

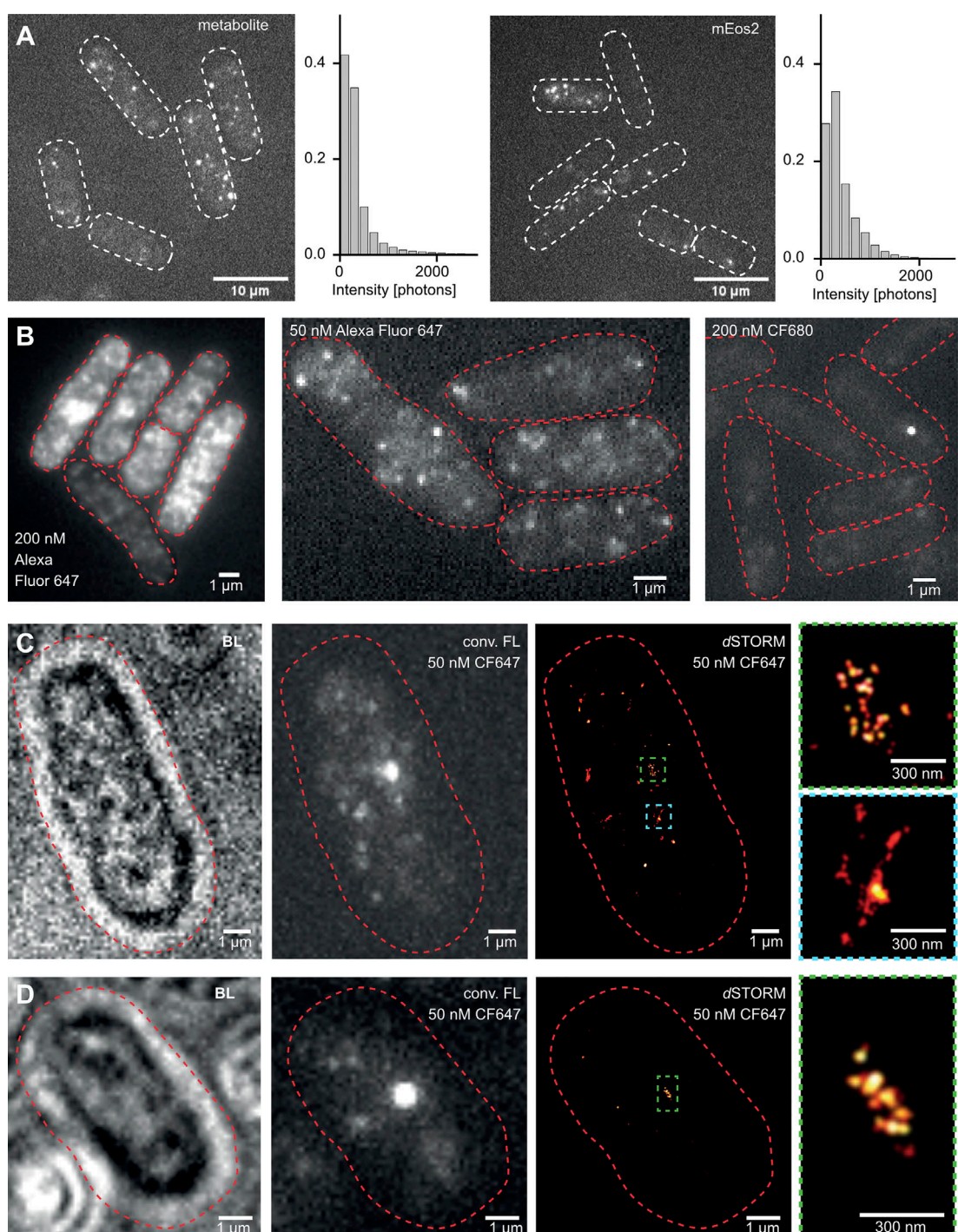

Figure S1.  **Developing the labeling strategy. (A)** Autofluorescence of metabolites overlaps with fluorescent protein signal intensity. Exemplary images from recorded movies of *S. pombe* cells containing either accumulated metabolites (due to a mutation in the ade6 gene, left) or expressing fluorescent proteins (cytosolic mEos2, right) at similar imaging conditions. Cell borders are shown as dashed lines. Scale bar, 10 µm. Histograms of the intensities of individual localizations for both conditions. Localizations were filtered using a sigma of 70–200 nm. The metabolite signal strength overlaps with the fluorophore signal. Therefore, the autofluorescence noise cannot be reliably filtered from the fluorophore signal. Metabolite N = 35,509, mEos2 N = 467,680, bin size = 200 nm. **(B)** Dye staining in *S. pombe* is heterogeneous and unreliable for low copy number targets. Staining of Halo-cnp1[CENP-A] cells with the cyanine Alexa Fluor 647 200 nM (left), 50 nM (middle) shows a high degree of nonspecific labeling, even when using the blocking agents BSA and Image-IT (which reduce nonspecific interactions of the charged dye with the sample). Staining with 200 nM (right) of cyanine CF680 shows low labeling efficiency even after cell wall and membrane were partially digested with zymolyase and Triton-X100. We attribute the low efficiency to its large molecular weight (CF680 is about 2.3 times larger than Alexa Fluor 680 as it has masking groups to avoid nonspecific staining due to charges). **(C)** Exemplary cell showing high unspecific staining of CF647-Halo-cnp1[CENP-A] in brightlight (left), conventional fluorescence (middle), and SMLM imaging (right). Detailed views illustrating inset with the cnp1[CENP-A] signal (green border) compared to some nonspecific signal (inset with blue border). **(D)** Exemplary cell with low, nonspecific staining of CF647-Halo-cnp1[CENP-A] at a high labeling efficiency.

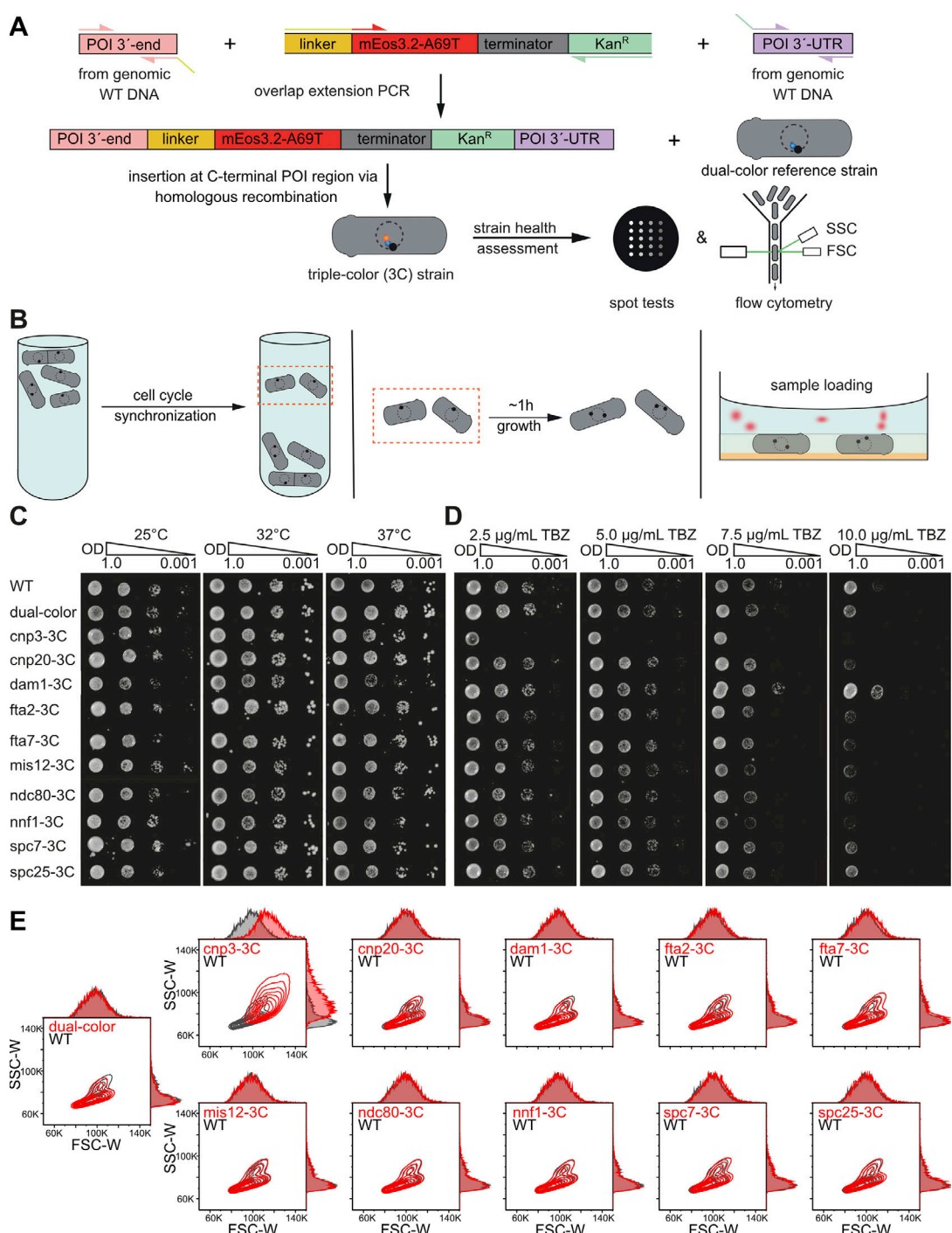

Figure S2. **Cloning strategy, sample preparation, and health assessment of 3C strains. (A)** Cloning strategy. DNA fragments containing the POI 3'-end (pink), the FP-resistance cassette (yellow-red-grey-green), and the POI 3'-UTR (purple) were amplified with the corresponding primers with ~20 bp overlap to the future neighboring DNA fragments from either genomic WT DNA or a plasmid DNA. Pieces were then fused by overlap extension PCR and transformed into the dual-color reference strain (h+, leu1-32, ura4-D18, sad1:mScarlet-I:hphMX6, PAmCherry1:cnp1CENP-A) using homologous recombination to create a 3C strain library (Table S1). **(B)** Sample preparation. Cell cultures were synchronized using lactose gradient centrifugation, which accumulates cells in early G2 phase in an upper band. The cells were then extracted from the gradient column, grown for another 1–1.5 h until mitosis, and chemically fixed, washed, and embedded in agarose gel with fiducial markers. **(C and D)** Assessment of strain health & mitotic defects. Temperature (C) or TBZ (D), which induces mitotic defects by microtubule depolymerization, sensitivities between the parental WT, the dual-color reference, and the individual 3C strains were assessed by spot tests. Here, a 10-fold dilution series of OD$_{600}$ from 1.0 to 0.001 of overnight cultures was grown on either YES media plates at 25, 32, and 37°C for 3 d or YES media plates containing 2.5, 5.0, 7.5, and 10.0 µg/ml TBZ and incubation at 25°C for 3 d. **(E)** Flow cytometry measurements to assess *S. pombe* strain health. FSC-W and SSC-W contour plots (each level within the contour plot consists of 10% measured cells) and their corresponding histograms of 3C (POI-3C) strains and the dual-color template strain compared to a fission yeast WT strain. A defect in cell division usually results in increased cell length and higher FSC-W and SSC-W values, which we observed for cnp3CENP-C-3C (top row, left column) but none of the other strains tested. N = 10,000 for each POI (see Materials and methods).

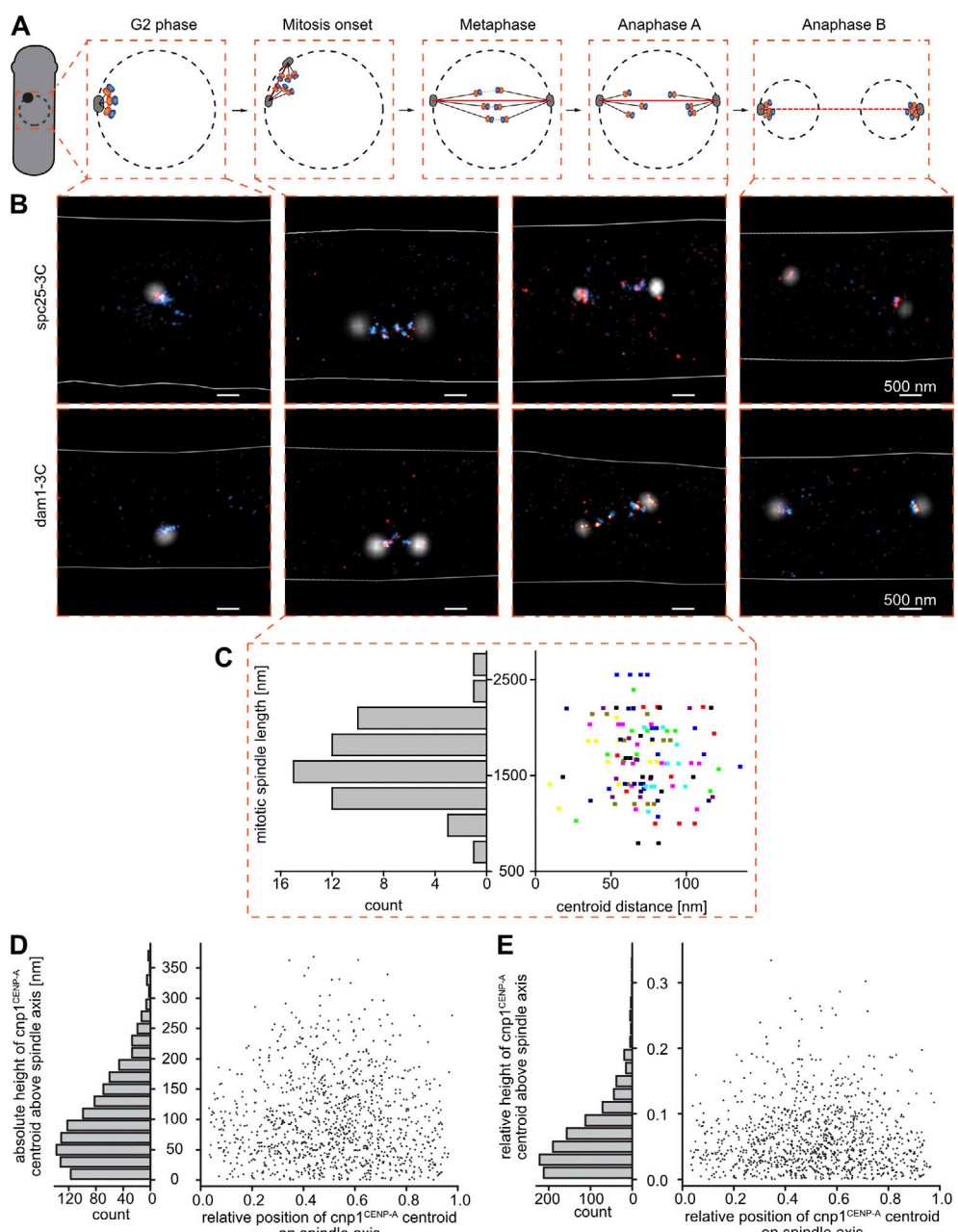

Figure S3.  **Centroid distance is independent of mitotic spindle length. (A)** Scheme of the *S. pombe* cell cycle (from left to right): A fission yeast cell in G2 phase is drawn with a nuclear envelope (black dashed circle) and one SPB (black). Insets show the nucleus changing over the cell-cycle phases (G2, onset of mitosis, metaphase, Anaphase A, and Anaphase B) with the KT (orange) linking the centromere (blue) to the SPB through a bundle of KT microtubules (black lines), while the spindle microtubules (red line) push the two SPBs further apart. **(B)** Exemplary three-color SMLM images of spc25-3C (top row) and dam1-3C (bottom row) strains representing the cell-cycle stages shown in A. SPB (sad1-mScarlet-I) localizations are shown in white, KTs (POI-mEos3.2-A69T) in red, centromeres (PAmCherry1-cnp1$^{CENP-A}$) in blue, and the cell border (determined by the bright light image) is drawn as a white line. Scale bar, 500 nm. **(C)** Left: Histogram of an exemplary dataset of mitotic spindle lengths (distance between the two SPBs during mitosis) for the POI dam1. Right: The distance between the centroids of individual KT cluster pairs (POI and cnp1$^{CENP-A}$) plotted against the mitotic spindle length from the same cell. Data points of the same height and color are from the same mitotic spindle. All spindle lengths are shorter than the average nuclear diameter of 2–3 µm, thus excluding Anaphase B cells. Mitotic spindles N = 55, KT cluster pairs N = 122, bin size = 280 nm. **(D and E)** Angular offset of kMT and spindle axes. Plotted are relative position and height over the spindle axis (defined as sad1-sad1 centroid distance) for all measured cnp1$^{CENP-A}$ centroids. Height of cnp1$^{CENP-A}$ centroids is either plotted in absolute nanometer distances (D) to visualize that most KTs are in direct vicinity to the central bundle or normalized to the respective spindle length of the cells (E) to represent the angular distribution between the spindle and kMT axes. N = 1,099, bin size = 20 nm.

Provided online are five tables and two datasets. Table S1 shows *S. pombe* and *E. coli* strains used in this study. Table S2 list of primers used for constructing the 3C-library and the FtnA protein standard as listed in Table S1. Table S3 shows weighted mean of localization counts for POIs belonging to the same KT subcomplex. Table S4 shows POI copy number ratios between different KT subcomplexes. Table S5 shows comparison of protein cluster distances between our study in *S. pombe* and the study of Cieslinski et al. (2023) in *S. cerevisiae*. Data S1 contains the STAN code and raw distance data for distance analysis. Data S2 contains two exemplary annotated kinetochore data sets and a short info.txt.

