## [Peer Review File · The Journal of Cell Biology]

Unraveling the kinetochore nanostructure in *Schizosaccharomyces pombe* using multi-color SMLM imaging

David Virant, Ilijana Vojnovic, Jannik Winkelmeier, Marc Endesfelder, Bartosz Turkowyd, David Lando, and Ulrike Endesfelder

Corresponding Author(s): Ulrike Endesfelder, MPI for Terrestrial Microbiology

Review Timeline:	Submission Date:	2022-09-24
	Editorial Decision:	2022-11-21
	Revision Received:	2022-12-20

Monitoring Editor: Arshad Desai

Scientific Editor: Dan Simon

Transaction Report:

DOI: <https://doi.org/10.1083/jcb.202209096>

Revision 0

Review #1

1. Evidence, reproducibility and clarity:

Evidence, reproducibility and clarity (Required)

Virant and colleagues have devised well-thought-out experimentation and analysis pipelines to obtain unbiased measurements of kinetochore protein counts and distances from the centromeric histone known as Cnp1 in fission yeast. My concerns with this study are mainly regarding the clarity of some of the data analysis strategies and data presentation. The authors should be able to address these concerns without new experimentation.

1. Segmentation of individual centromeres: In general, the authors are to be commended for including a detailed description of their procedures and analysis method. However, it wasn't readily clear to me exactly how they segmented individual centromeres. The lack of a consistent offset between the fluorescence spots corresponding to the protein of interest and Cnp1 in image in Figure 1 makes this issue even more confusing. It will help to display representative segmented individual kinetochores either in the main figure or a supplementary figure.
2. Use the mixture coefficient: The authors use the coefficient λ to create a mixture model for the Bayesian inference of distances. The description provided in the methods section is not sufficient for an average reader to understand how this coefficient is ultimately used (I had to look up the code and then the Stan manual for a superficial understanding of this procedure). It will be very helpful to flesh out this part of the model. Similarly, notation for the model that they use should be included either in the Methods or in supplementary data so that casual readers can get some understanding of the model.
3. The regional centromeres in fission yeast can incorporate varying levels of Cnp1 depending on its expression level (e.g. see Aravamudhan et al. 2013 Current Biology, Joglekar et al. 2008 JCB). Much of this "extra" Cnp1 is likely to be incorporated at sites distal to the Cnp1 molecules that directly nucleate the kinetochore. Therefore, the centroid of Cnp1 molecules is likely to be "shifted" to some extent from the foundation of the kinetochore. Any shift in the Cnp1 centroid will be important especially when comparing the fission yeast measurements with the budding yeast data. The authors should ascertain whether such a shift can be detected by comparing the budding yeast and fission yeast measurements.

2. Significance:

Significance (Required)

Virant and colleagues present a rigorous single-molecule localization-based analysis of the kinetochore protein copy number and organization within the fission yeast kinetochore. Although the fission yeast kinetochore has been extensively studied, the spatial organization of its

kinetochore components has remained uncharacterized. This manuscript addresses this deficiency, and in concert with the budding yeast study, highlights the conserved and diverged features of the kinetochore in the two yeast species. Therefore, this manuscript will be of great interest to the kinetochore and cell division field.

3. How much time do you estimate the authors will need to complete the suggested revisions:

Estimated time to Complete Revisions (Required)

(Decision Recommendation)

Less than 1 month

Review #2

1. Evidence, reproducibility and clarity:

Evidence, reproducibility and clarity (Required)

In this study, Virant and colleagues have applied single molecule localization microscopy to map the positions of proteins in the pombe kinetochore. This has not been reported previously and this study is both well-conducted and the data appear solid. They also use a modification of this technique to assess the stoichiometry of kinetochore proteins. The results that they obtain are broadly in line with several previous studies that use other methodology but not in fission yeast nor to this level of detail. There are some important novel conclusions from this work. I would like the authors to address the following concerns prior to publication:

1. It is not clear to me why the sad1-Scarlett-I signal in Figure 1C, displays a grid-like pattern? This must be an artefact of image collection or processing. Could the authors explain this pattern since this may affect the ability to find a centroid position of this signal?
2. It is my understanding that the distances reported are based on the positions of the proteins in one dimension, along the spindle axis, consistent with other studies and as illustrated in Figure 1b. This should be clearly stated in the results section.
3. The distances between proteins in this study are measured during anaphase, whereas the distances measured in cerevisiae previously were in both metaphase and anaphase (Joglekar et al 2009) and in the accompanying manuscript (Cielinski et al) in metaphase. In the comparison of distances, it would be worth describing how the mitotic stage may have affected distances, since Joglekar et al, found significant positional changes in cerevisiae kinetochore proteins from metaphase to anaphase.
4. It is hard to interpret the POI copy numbers in terms of each kMT. I am assuming that each cluster measured represents a single pombe kinetochore, containing 2-4 kMTs? If we assume that

each pombe kinetochore can contain 2, 3, or 4 kMTs, then we might expect to see a trimodal dataset, I am guessing this was not seen in the data? Would it be possible to estimate protein numbers per kMT in Table 2, as done for the Cielinski et al study? I realise this would require an estimate of the number of kMTs per kinetochore. Alternatively, the authors are resolving individual kMTs, in which case this should be made clear.

5. The same kMT issue may affect the measurements of distances. Each pombe kinetochore contains multiple kMTs and it is not clear whether these would align perfectly on the spindle axis. Did the authors see anything in their data that would support the notion that individual kMTs are aligned on the axis (as illustrated in Figure 2) or whether they are slightly separated? This is itself a potentially important result.

6. In all measurements of kinetochore protein intensity (both in this study and previous studies) there seems to be significant variation in the data for individual kinetochores, even for *S. cerevisiae*, which supposedly has a fixed number of the kMTs. The coefficient of variation is ~ 0.5 in the data shown in Table 2. Could the authors discuss the variability in POI copy numbers since it either reflects an inability to measure protein levels accurately or that there is some flexibility in kinetochore protein stoichiometry (or in this case differing numbers of kMTs per kinetochore - see point above)?

****Minor points:****

Delete "an" from "...structure at an about 100 nm resolution" (page 3).

In Figure 2 the proteins in the schematic are color coded, but it is not clear what the coloured proteins are in all cases. Would it be possible to color code the adjacent text, e.g. Spc7 in orange. Also in this figure, the POI copy numbers are indicated by color coding of the data points. However, the points will likely be too small in the final figure for these colors to be clearly visible. Perhaps copy numbers could be indicated in another way or the "mean value" boxes could be larger?

Please define "N" in Table 2. e.g. N = number of kinetochores measured.

2. Significance:

Significance (Required)

This manuscript, together with an accompanying one from Cielinski et al., are nice complementary studies that provide the first single molecule localization studies of the yeast kinetochore. Although other labs have used super-resolution methods to study individual kinetochore proteins; both of these new studies map distances between many proteins at the kinetochore and thus are able to produce maps of the overall kinetochore structure. Like the previous study using standard resolution methods (Joglekar et al, 2009. Current Biology 19, 694-699); these studies will likely provide a benchmark for future studies on eukaryotic kinetochore architecture, including those in mammalian systems. Additionally, this work will appeal to super-resolution microscopists.

My expertise is as a yeast kinetochore cell biologist.

3. How much time do you estimate the authors will need to complete the suggested revisions:

Estimated time to Complete Revisions (Required)

(Decision Recommendation)

Between 1 and 3 months

Manuscript number: RC-2021-01180

Corresponding author(s): Ulrike, Endesfelder

1. General Statements [optional]

This section is optional. Insert here any general statements you wish to make about the goal of the study or about the reviews.

This is a back-to-back submission and parallel revision with a budding yeast manuscript the Ries group (RC-2021-01179). As the reviewers kindly reviewed both manuscripts, we quickly would like to give the general overview over both revisions before going into details of our own revision in 2. We explicitly addressed all comments of both manuscripts:

- New experimental data: the accompanying Ries manuscript now added data for the DASH ring protein ask1. This new data aligns with our dam1 data as we propose that *S. pombe* the c-terminal region of dam1 actually localizes at the DASH ring and not at the ndc80 head domain (as e.g. in *S. cerevisiae*). The *S. cerevisiae* ask1 data marks the position of the DASH ring. The new distances of the Ries group fit our measured distances of *S. pombe* dam1. This new data thus strongly supports the proposed positioning of *S. pombe* dam1 in the DASH ring.
- Extended data analyses in both manuscripts including estimating measurement errors, re-checking all distances and protein copy numbers, and adding the auto-correlation analysis of protein distributions in the Ries lab, and analyzing the angular distribution between mitotic kMT axes and spindle axes for *S. Pombe* in the Endesfelder lab manuscript.
- Extended discussions and explanations. For the *S. pombe* manuscript this entails: adding exemplary data sets to illustrate the analysis procedures in more details (segmented clusters, exemplary data of inner POIs), commenting of the stan code to improve the readability of the code and new SI figure to visualize the Bayesian model, adding background info for the reviewers including preliminary images on microtubule atb2 labeling to illustrate the mitotic microtubule bundles, adding discussions (e.g. using the variance of the FtnA data as a proxy for the technical error in POI counting, influence of angle between kMT axes and spindle axes). For the *S. cerevisiae* manuscript this entails: clarifying how the analyses were performed, more extensive discussions of the data, discussion of different models for the cse4 copy numbers, and including discussions on previous works to better relate our findings to the state-of-the art in the field.

This section is mandatory. Please insert a point-by-point reply describing the revisions that were already carried out and included in the transferred manuscript.

Reviewers comments are in black, our answers in blue.

Virant and colleagues have devised well-thought-out experimentation and analysis pipelines to obtain unbiased measurements of kinetochore protein counts and distances from the centromeric histone known as Cnp1 in fission yeast. My concerns with this study are mainly regarding the clarity of some of the data analysis strategies and data presentation. The authors should be able to address these concerns without new experimentation.

We would like to thank reviewer 1 for their valuable comments. We have addressed all comments and highlighted the changed sections in our revised manuscript.

1. Segmentation of individual centromeres: In general, the authors are to be commended for including a detailed description of their procedures and analysis method. However, it wasn't readily clear to me exactly how they segmented individual centromeres. The lack of a consistent offset between the fluorescence spots corresponding to the protein of interest and Cnp1 in image in Figure 1 makes this issue even more confusing. It will help to display representative segmented individual kinetochores either in the main figure or a supplementary figure.

We thank reviewer #1 for this comment and highly appreciate that they value our effort for detailed method descriptions. We strongly agree: Correct segmentation and best-possible visualization are crucial for our analyses. We hope that the following clarifications help to understand our work better:

All images of the manuscript are reconstructed from localization files using Rapidstorm, which linearly interpolates the localizations on a subpixel grid and fills the pixels based on the distance between the localization and the center of the main subpixel bin to avoid discretization errors (Wolter et al., 2012). These images are then overlaid with a Gaussian blur corresponding to their localization uncertainty. In our opinion, this procedure gives the most realistic image impression of the data and results in reconstructed images that as best as possible mimic real fluorescence images. This is in our opinion a very important aspect when presenting SMLM data in reconstructed images as it is crucial that one - when looking at those images - does not over- or under-interpret the SMLM data. We extended the explanation in the manuscript: *“For visualization, we aimed to reconstruct SMLM images that neither over- nor under-interpret the resolution of the SMLM data and resemble fluorescence images as closely as possible. Localizations were tracked together using the Kalman tracking filter in Rapidstorm 3.2 with two sigma, and the NeNA value used as sigma. Images were then reconstructed in Rapidstorm 3.2 with a pixel size of 10 nm. Rapidstorm linearly interpolates the localizations and fills neighboring pixels based on the distance between the localization and the center of the main subpixel bin to avoid discretization errors (Wolter et al., 2012). These images were then processed with a Gaussian blur filter based on their NeNA localization uncertainty in the open-source software ImageJ 1.52p (Schindelin et al., 2012). Importantly, images were only used for image representation purposes, all data analysis steps were conducted on the localization data directly (see data analysis).”*

Importantly, individual centromeres were segmented not on the images but on the localization data directly which we visualized in a self-written 3D visualization software that has the functionality to e.g. zoom in and out and to switch or overlay channels. This flexible visualization tool allowed us to make the best-possible informed decisions for the cluster selections and pairing of POI/cnp1^{CENP-A} pairs. We extended our explanation by: *“For several manual steps, localizations were visualized in a custom software, which allows to zoom in/out flexibly and to switch/overlay between the sad1/POI/ cnp1^{CENP-A} channels. Using this tool, individual localizations could be*

selected and classified. For channel alignment, localizations belonging to the same fiducial marker in all three channels were grouped together. Cells with visible kinetochore protein clusters in the focal plane were selected and classified as individual region of interests (ROIs) and all clusters were annotated. $cnp1^{CENP-A}$ clusters were paired together with corresponding POI clusters. Whenever there was any doubt whether two clusters belonged to the same kinetochore or whether a cluster represented a single centromere region or several, the clusters were discarded. Two exemplary data sets can be found in the zip-file Supplementary Data 1. The annotation work was quality-checked by cross-checking the annotation of two different persons.”

Maybe interesting to add is that we initially and extensively tried several ways to fully automate the annotation. As all tested routines could not reach the quality of manually annotated data and had to a large extent be manually rechecked and corrected, we in the end directly annotated the data manually. We were rigorous in case of doubt: “Whenever there was any doubt whether two clusters belonged to the same kinetochore or whether a cluster represented a single centromere region or several, the clusters were discarded (manuscript page 10, section Visualization and Manual Analytics).”

Furthermore, the reviewer is right, we didn't include too much visual data/results into our manuscript. We now showcase some examples for our annotation. In the zip-folder “Supplementary Data 1”, the reviewer will find two csv SMLM data examples of annotated localization data of cells with a mitotic spindle that passed the quality checks of drift control and channel overlay. For the first example, a cell with $dam1$ as POI, all 6 kinetochores are visible and all were grouped and separated from noise. One can also nicely see (due to the large distance of $dam1$ to $cnp1^{CENP-A}$) which of the six belong to which spindle by the spatial orientation of the $cnp1^{CENP-A}$ and $dam1$ cluster to each other, and both groups of three have a pair that is spatially closer to the wrong pole, important for question 2 below. The second example, with $spc7$ as the example POI, has only 5 visible clusters. A closer look shows that one of them has unusual dimensions and a high number of localizations, representing most likely two overlapping centromeric regions. Thus, for this cell, only 4 kinetochore pairs were annotated for further analysis. Also, the cluster pairs are overlapping much more, so a direct decision to which spindle they belong to gets difficult.

Finally, we realized that the example cell used for figure 1 which we chose a long time ago for representative purposes actually is a dataset that did not pass the final quality controls of drift correction and channel overlay. Thus, it is not part of the data that was used for the results of this work. While it is pretty embarrassing that we did not realize earlier, we are really grateful for the reviewers' question about it. We now replaced it by another example which nicely represents the data that went into the analysis and also represents the biological heterogeneity, not only in offset but also e.g. in shape: In this work, we simplify by ignoring any shape in our current analysis and only use cluster centroid distances. Kinetochore POI cluster shapes are currently investigated in a more detailed follow-up study.

2. Use the mixture coefficient: The authors use the coefficient λ to create a mixture model for the Bayesian inference of distances. The description provided in the methods section is not sufficient for an average reader to understand how this coefficient is ultimately used (I had to look up the code and then the Stan manual for a superficial understanding of this procedure). It will be very helpful to flesh out this part of the model. Similarly, notation for the model that they use should be included either in the Methods or in supplementary data so that casual readers can get some understanding of the model.

We added the following sentence to explain the significance of the mixture coefficient:

"The coefficient then corresponds to the prior probability that the centromere is attached to the first spindle pole."

We are not sure whether we understand the last sentence of this comment correctly. We added more explanations and definitions to the Stan code (see kinetochore.stan) to make it easier to understand. However, the section "Distance calculation" is intended to give the reader a full understanding of all relevant parts of the model, a look at the code should therefore not be necessary. The Stan implementation of the model uses non-centered parameterization, which might make it appear more complex than what is described in the text. However, this is an implementation detail that is intended to make the posterior more well-suited for Monte Carlo estimation and does not change the underlying statistical model. It should therefore be of no concern to casual readers. Finally, we prepared a new Supplementary Figure S6 to visualize the model for all readers.

3. The regional centromeres in fission yeast can incorporate varying levels of Cnp1 depending on its expression level (e.g. see Aravamudhan et al. 2013 Current Biology, Joglekar et al. 2008 JCB). Much of this "extra" Cnp1 is likely to be incorporated at sites distal to the Cnp1 molecules that directly nucleate the kinetochore. Therefore, the centroid of Cnp1 molecules is likely to be "shifted" to some extent from the foundation of the kinetochore. Any shift in the Cnp1 centroid will be important especially when comparing the fission yeast measurements with the budding yeast data. The authors should ascertain whether such a shift can be detected by comparing the budding yeast and fission yeast measurements.

The reviewer is absolutely right, there is an active discussion in the field to which extend there are *cnp1^{CENP-A}* molecules at distal sites and thus not part of the platform for the kinetochore structure. While different quantification methods in the literature partly disagree in *cnp1^{CENP-A}* numbers, we have no indications that our own assessment by PALM imaging is wrong. In this study, we get a very stable (Suppl. Figure S9 b) *cnp1^{CENP-A}* read-out by PamCherry1, which agrees with our Lando et al. 2012 study (a single color study with a different, much simpler protocol and using a different microscope with different settings and analysis routines, mainly using mEos2 but also some SI data on PamCherry1 for counting *cnp1^{CENP-A}* molecules). Furthermore, the recombinant fusions in the native locus of *cnp1^{CENP-A}* are stable and the strains show no signs of growth or phenotypic defects.

While we therefore can argue that we see a native level and undisturbed distribution of *cnp1^{CENP-A}*, we nevertheless do not know how much of this *cnp1^{CENP-A}* is involved in building up the kinetochore. What we believe to know is the following:

- i) With our ChipSeq data in Lando et al. 2012 we explored the distribution and read-out hits of *cnp1^{CENP-A}* within the outer repeats as well as the inner centromeric region of all three chromosomes (Figure 3 and SI in Lando et al. 2012). While *cnp1^{CENP-A}* is highly populated within the ~ 10 to 15 kb large inner centromeric regions, there are less detections in the outer regions. Thus, while ChipSeq is not a quantitative method, we believe it's showing the correct trend with some *cnp1^{CENP-A}* in the outside regions but most *cnp1^{CENP-A}* localizing in the inner region. We believe that in overexpression studies (like e.g. done both in Joglekar et al. 2008 & Aravamudhan et al. 2013), this most likely will differ (but we did not experimentally explore this).
- ii) We generally would argue that our distances are rather accurate using a symmetry and compaction argument. The about 10kb inner regions are a roughly 3.4 μm long DNA strands at a linear 1-dimensional scale but *in vivo* are highly compacted. Total kinetochore sizes as seen in EM data for mammalian cells are "approximately 250 nm wide and 80 nm deep, with an electron-opaque inner plate juxtaposed to the centromeric chromatin, a

translucent gap layer, and an electron-opaque, chromatin-distal outer plate apparently embedding the plus ends of spindle microtubules” (Musacchio and Desai, *Biology* 2017, 6(1), 5; 10.3390/biology6010005) and for *S. pombe* also in the 200 nm range (Ding et al., *J Cell Biol* (1993) 120 (1); 10.1083/jcb.120.1.141). We evaluated the *cnp1*^{CENP-A} cluster shapes as seen in our SMLM data. The clusters show a major axis length of 218 ± 88 nm and a minor axis length of 110 ± 29 nm on average. Taking into account our NeNA localization precisions, this is in nice agreement with the EM data measuring the lateral extend of the kinetochore structure.

All together, we would argue a) that there is no reason or any indication in the literature that *cnp1*^{CENP-A} not directly involved in kinetochore nucleation preferably gets incorporated on only one of the distal sites and thus would cause an asymmetry. We rather would argue that they are randomly inserted on both sides at low level and thus keep the symmetry needed to determine the center of the *cnp1*^{CENP-A} cluster involved as the kinetochore platform. We also would argue b) that the structure is highly compacted and thus errors caused by additional *cnp1*^{CENP-A} molecules will be small in respect to our resolution. We cannot completely exclude that there is such an effect that would increase the measured distances, however, and given all other sources of error (drift correction, channel alignment, cluster selection etc.) this most likely is not the main factor in defining the widths of the posterior distributions as we obtain them (Suppl. Figure S7). This argument is supported by the fact that we do not see any indication of such an effect for *cnp1*^{CENP-A}-close proteins. We carefully checked *fta2*, *fta7* and *cnp20* data and also included some examples for the reviewer (see innerPOlexamples.zip). We hope that the reviewer agrees with us that there is no indication for a systematic asymmetric offset between the POI and *cnp1*^{CENP-A} clusters. Finally, our distance numbers nicely agree with the distances the Ries group has measured for *S. cerevisiae* in the co-submitted manuscript. They a) have a point centromere with presumably only one kMT and b) did not use *cnp1*^{CENP-A} as their reference, they used *spc7*^{KNL1} (*spc105*).

Reviewer #1 (Significance (Required)):

Virant and colleagues present a rigorous single-molecule localization-based analysis of the kinetochore protein copy number and organization within the fission yeast kinetochore. Although the fission yeast kinetochore has been extensively studied, the spatial organization of its kinetochore components has remained uncharacterized. This manuscript addresses this deficiency, and in concert with the budding yeast study, highlights the conserved and diverged features of the kinetochore in the two yeast species. Therefore, this manuscript will be of great interest to the kinetochore and cell division field.

We would like to thank the reviewer again for their very helpful and highly constructive review.

Reviewer #2 (Evidence, reproducibility and clarity (Required)):

In this study, Virant and colleagues have applied single molecule localization microscopy to map the positions of proteins in the pombe kinetochore. This has not been reported previously and this study is both well-conducted and the data appear solid. They also use a modification of this technique to assess the stoichiometry of kinetochore proteins. The results that they obtain are broadly in line with several previous studies that use other methodology but not in fission yeast

Full Revision

nor to this level of detail. There are some important novel conclusions from this work. I would like the authors to address the following concerns prior to publication:

We would like to thank Reviewer #2 for their appreciation of our work and their helpful remarks regarding our manuscript which we will answer below.

1. It is not clear to me why the *sad1*-Scarlet-I signal in Figure 1C, displays a grid-like pattern? This must be an artefact of image collection or processing. Could the authors explain this pattern since this may affect the ability to find a centroid position of this signal?

Thanks a lot for this comment. Yes, the grid-like pattern the reviewer observed is an artefact from image processing when compiling the exemplary composite 3-color images. We revisited the raw movie data and changed the procedure to produce the exemplary images to avoid this ugly artifact (which did not influence our data analysis as it is not present in the raw movies used for centroid determinations). Please note: we also changed the example cell for figure 1 due to reasons explained in the answer 1 to reviewer 1.

2. It is my understanding that the distances reported are based on the positions of the proteins in one dimension, along the spindle axis, consistent with other studies and as illustrated in Figure 1b. This should be clearly stated in the results section.

The model underlying our Bayesian inference is that *cnp1*^{CENP-A}-POI and one of the two *sad1* spindles are all on one “mitotic” axis, along the kinetochore microtubule. BUT the orientation of this mitotic axis is NOT necessarily parallel to the spindle axis. See Figure 1a, the in red drawn spindle axis is not necessarily parallel to the green drawn microtubules connecting the kinetochores to the spindle poles. (Please note, thanks to this comment, we found that our original sketch of Figure 1a was misleading and corrected for that.). Figure 1b in this respect is a bit misleading as only one spindle pole is shown. The slight difference between the kinetochore and spindle axis cannot be visualized with only half a spindle. For answering the reviewers comment no. 5, we also now plotted the measured offset from and relative position of *cnp1*^{CENP-A} cluster centroids to the spindle axis, see below.

3. The distances between proteins in this study are measured during anaphase, whereas the distances measured in cerevisiae previously were in both metaphase and anaphase (Joglekar et al 2009) and in the accompanying manuscript (Cielinski et al) in metaphase. In the comparison of distances, it would be worth describing how the mitotic stage may have affected distances, since Joglekar et al, found significant positional changes in cerevisiae kinetochore proteins from metaphase to anaphase.

We would like to thank the reviewer for this comment. It sparked a longer discussion to precisely disentangle the cell cycle states. We afterwards went carefully through the literature again and added a column to Table S4 stating for which cell cycle phase(s) the individual works were conducted which we believe is highly useful when comparing the different data.

Discussing with the Ries group we made sure that we indeed measured the cells in the same state and that we in our manuscript made mistakes in defining it correctly. Important for *S. pombe* is, that while it is not possible to decide for 100% between metaphase and anaphase A, we can safely exclude anaphase B so that we can state that we did not image anaphase B: Supplementary Figure S10 shows that all spindle distances measured are smaller than one

nuclear diameter which is given by 2-3 μm (MacLean 1964, 10.1128/jb.88.5.1459-1466.1964; Tda et al 1981, 10.1242/jcs.52.1.271) plus we ourselves measured a nup132 strain in early G2 phase and obtained $2.4 \mu\text{m} \pm 0.19 \mu\text{m}$ (data not shown)). This is nicely in line with Joglekar et al. 2009 (this paper actually has a very good SI figure on exactly this topic, see Fig. S1) and corrected the manuscript accordingly. Thanks for pointing this out to remove this lapse in definitions.

4. It is hard to interpret the POI copy numbers in terms of each kMT. I am assuming that each cluster measured represents a single pombe kinetochore, containing 2-4 kMTs? If we assume that each pombe kinetochore can contain 2, 3, or 4 kMTs, then we might expect to see a trimodal dataset, I am guessing this was not seen in the data? Would it be possible to estimate protein numbers per kMT in Table 2, as done for the Cielinski et al study? I realize this would require an estimate of the number of kMTs per kinetochore. Alternatively, the authors are resolving individual kMTs, in which case this should be made clear.

Yes, the reviewer is absolutely correct, the clusters are associated with 2-4 kMT (as nicely resolved in Ding et al, J Cell Biol (1993) 120 (1); 10.1083/jcb.120.1.141). Thus, we can assume that 2-4 kinetochore structures are also involved per centromeric region. In our current analysis, we have to work with the average of 2-4 kMTs. The shapes of POI and *cnp1*^{CENP-A} clusters we have in the SMLM data are definitely diverse, and we plan to extract more data on spatial distribution in the future, perhaps even at the level of individual centromeric regions, but we did not systematically explore shapes in this work. Thus currently, we cannot give a precise answer how individual regional kinetochores look like at the level of a single kinetochore, but we strongly agree with the reviewer that this will be highly interesting to explore further. In this manuscript, therefore, to compare the stoichiometries between the point centromere of *S. cerevisiae* and the regional centromere of *S. pombe*, we used ratios as given in Suppl. Table S4. These ratios provided us with comparable results across a wide range of literature since the ratios are only calculated internally for each study and not across studies (which would lead to compatibility issues).

For this study, we also labeled MT via atb2. Unfortunately, the SMLM experiments were very difficult as atb2 is also present everywhere else in the nucleus, in particular at the dense central MT bundle (see image below, white sad1-mScarlet-l, blue PamCherry1- *cnp1*^{CENP-A}, and red mEos3.2-A69T-atb2). Thus, we could not resolve such fine details as single fibers for the kMTs: Most kMTs overlapped with the central fiber and due to the dense central MT bundle of atb2, the data of the atb2 channel could not be read-out neither in a quantitative nor complete way and we could not extract which percentage of atb2 molecules we actually successfully recorded in the SMLM data. Thus, especially visualizing fine fibers was difficult and the images obtained do not meet our quality standards – but the exemplary one below is maybe nevertheless informative in a qualitative way. Our current idea would be to use ExM plus SMLM, but this work would be a stand-alone study requiring the set-up and optimization of such a protocol.

5. The same kMT issue may affect the measurements of distances. Each pombe kinetochore contains multiple kMTs and it is not clear whether these would align perfectly on the spindle axis. Did the authors see anything in their data that would support the notion that individual kMTs are aligned on the axis (as illustrated in Figure 2) or whether they are slightly separated? This is itself a potentially important result.

It is important to note that we did not measure the distance in projection of the spindle axis (defined as *sad1-sad1* centroid axis), see also question 2. We can show in our data that the main microtubule bundle between the spindles is angled to the kinetochore microtubules that connect the centroids of our three-color channels for each centromeric region in a *sad1-POI-cnp1* axis. For *sad1*, we cannot simply determine which one of the two spindles is the correct one, thus we implemented a mixture model for our Bayesian model, see also answers to reviewer 1).

While in the budding yeast literature it has been measured by EM that there are only small angles of up to 6° between the two axes present (Joglekar et al. 2009, 10.1016/j.cub.2009.02.056, Figure S1), Ding et al. 1993 showed larger deviations for *S. pombe*. Our data agrees with these findings. In the new Suppl. Figure S12, we plotted the height of all measured *cnp1*^{CENP-A} centroids, normalized to the spindle lengths to represent the angular distribution between the spindle and kMT axes and in absolute nanometer distances to show that most kinetochores are in direct vicinity to the central bundle and only few show heights larger than 150 nm (also technically important as our focal z-range is ~600 nm).

6. In all measurements of kinetochore protein intensity (both in this study and previous studies) there seems to be significant variation in the data for individual kinetochores, even for *S. cerevisiae*, which supposedly has a fixed number of the kMTs. The coefficient of variation is ~ 0.5 in the data shown in Table 2. Could the authors discuss the variability in POI copy numbers since it either reflects an inability to measure protein levels accurately or that there is some flexibility in kinetochore protein stoichiometry (or in this case differing numbers of kMTs per kinetochore - see point above)?

Regarding the variability in our counting data we indeed expect a mixture of biological and technical nature in line with what the reviewer argues above but intuitively would lean towards the latter, being technically limited by the read-out precision of quantitative PALM imaging using fluorescent proteins, which have finite maturation- and read-out efficiencies and possess limited signal-to-noise contrast. When discussing this comment and how we possibly could support our

intuition by evidence, we realized that the main argument to be made is that the FtnA oligomer is a biologically highly defined structure and that the variance we obtain for our FtnA calibration standard indeed also directly can serve as a proxy for our technical variance. Using the results of 21.63 counts \pm 10.2 STD for the 24mers and of 7.27 counts \pm 2.72 STD for the 8mers, we thus can estimate that the technical variance causes a coefficient of variance of 0.35 to 0.5, thus almost completely explaining the experimentally seen coefficient of \sim 0.5. we added this argument to our manuscript by *"The POI protein copy numbers as given in Table 2 show a large coefficient of variation. To assess to which extent this variability reflects a technical inability to measure protein levels accurately or some flexibility in kinetochore protein stoichiometry (e.g. due to differing numbers of kMTs per kinetochore), we can use the data of the FtnA oligomer counting standard: The FtnA oligomer is a biologically highly defined structure. Thus, our FtnA measurements can directly serve as a proxy for the contribution of the technical inaccuracy of our PALM imaging and analysis strategy to the variance. Using the results of 21.63 counts \pm 10.2 STD for the 24mers and of 7.27 counts \pm 2.72 STD for the 8mers, we can estimate that the technical inaccuracy causes a coefficient of variance of 0.35 to 0.5, thus almost completely explaining the experimentally seen coefficient of \sim 0.5 for our POI data (Table 2). Due to this high technical inaccuracy, we cannot resolve sub-populations of possibly different kinetochore structures (and thus POI copy numbers) on 2-4 kMTs in our current counting data (Supplementary Figure S9)."* Secondly, as the reviewer also pointed out earlier, we have a biological variance of 2-4 kMTs and thus must assume smaller and larger kinetochores on individual centromeres. Due to the high technical variance, we nevertheless cannot resolve sub-populations in our current counting data (please also compare to the data in Supplementary Figure S9).

Minor points:

Delete "an" from "...structure at an about 100 nm resolution" (page 3).

Thanks!

In Figure 2 the proteins in the schematic are color coded, but it is not clear what the coloured proteins are in all cases. Would it be possible to color code the adjacent text, e.g. Spc7 in orange.

Thanks for this suggestion, adjusted figure accordingly.

Also in this figure, the POI copy numbers are indicated by color coding of the data points. However, the points will likely be too small in the final figure for these colors to be clearly visible. Perhaps copy numbers could be indicated in another way or the "mean value" boxes could be larger?

We tested several options and decided to adjust the widths of the boxes.

Please define "N" in Table 2. e.g. N = number of kinetochores measured.

We added this information to the caption: *"N number of centromeric regions analyzed."*

Full Revision

Reviewer #2 (Significance (Required)):

This manuscript, together with an accompanying one from Cielinski et al., are nice complementary studies that provide the first single molecule localization studies of the yeast kinetochore. Although other labs have used super-resolution methods to study individual kinetochore proteins; both of these new studies map distances between many proteins at the kinetochore and thus are able to produce maps of the overall kinetochore structure. Like the previous study using standard resolution methods (Joglekar et al, 2009. Current Biology 19, 694-699); these studies will likely provide a benchmark for future studies on eukaryotic kinetochore architecture, including those in mammalian systems. Additionally, this work will appeal to super-resolution microscopists.

My expertise is as a yeast kinetochore cell biologist.

We would like to thank Reviewer #2 again for their appreciation of our work and their valuable remarks and discussion points which improved our manuscript substantially during the revision phase.

Additional comments for both reviewers:

As the co-submitted manuscript from Cielinski et al. 2021 re-analyzed all their distances and numbers during the revision phase, we updated the comparisons in Suppl. Table S4, S5 and our summary text: 1) their *cnn1* distance measurement got corrected and now shows no deviation anymore to our data. Also, their protein copy numbers changed slightly. So we changed the summary from *“Additionally, two organism-specific differences surfaced: $cnp20^{CENP-T}$ (*cnn1*) is located between spindle pole and $cnp1^{CENP-A}$ in our case (at similar distance as $fta2^{CENP-P}$ and $fta7^{CENP-Q}$), whereas Cielinski et al. position *cnn1* (and *mif2*) behind $cnp1^{CENP-A}$. Furthermore, the ratio of $cnp20^{CENP-T}$ to COMA is 1:0.9 in our case and 1:2.1 for *S. cerevisiae*”* to *“Importantly, one substantial organism-specific difference for the inner kinetochore strategy surfaced: The ratio of $cnp20^{CENP-T}$ to COMA is 1:0.9 in our case and 1:2.0 for *S. cerevisiae*.”* 2) Additionally, we were able to add a discussion about their new measurement of *ask1* of the *dam1* complex. *ask1* is a protein of the DASH ring. Their new distance measurement of *S. cerevisiae ask1* fits to the distance we measure for *S. pombe dam1* and thus supports our discussion that, for *S. pombe*, the C-terminus of *dam1* localizes at the DASH ring and not at the *ndc80* heads like for *S. cerevisiae*. We added this sentence to the summary: *“Furthermore, the *S. cerevisiae* work measured the position of *ask1*, a protein of the DASH ring. Their positioning of *S. cerevisiae ask1* is consistent with the distance we measured for *S. pombe dam1* and thus directly supports our reasoning that the C-terminus of *S. pombe dam1* is localized to the DASH ring and not to the *ndc80* heads (like for *S. cerevisiae*).”*

We found that we – very unfortunately - did a calculation mistake ourselves and used the inverse of the correction factor (multiplied with 0.9 instead of dividing with 0.9) when correcting our SMLM localizations to absolute protein counts. Thus, the numbers we gave in Table 2 and the color bar in Figure 2 were wrongly converted. We now corrected for this lapse.

November 21, 2022

RE: JCB Manuscript #202209096T

Prof. Ulrike Endesfelder
MPI for Terrestrial Microbiology
Department of Systems and Synthetic Microbiology
Karl-von-Frisch-Str. 16
Marburg 35043
Germany

Dear Prof. Endesfelder,

Thank you for submitting your revised manuscript entitled "Unraveling the kinetochore nanostructure in *Schizosaccharomyces pombe* using multi-color single-molecule localization microscopy." We would be happy to publish your paper in JCB pending the minor textual changes recommended by the reviewers as well as final revisions necessary to meet our formatting guidelines (see details below).

A. MANUSCRIPT ORGANIZATION AND FORMATTING:

1) Text limits: Character count for Reports is < 20,000, not including spaces and there should be a single combined 'Results and Discussion' section. Count includes title page, abstract, introduction, results, discussion, and acknowledgments. Count does not include materials and methods, figure legends, references, tables, or supplemental legends.

2) Figure formatting: Reports may have up to 5 main text figures and are generally limited to 3 supplemental figures. We will be able to give you extra space for supplemental materials but since you currently only have 2 main figures we ask that you please move some of the supplemental data to the main text. We think that Figures S7, S9, S10, and maybe S12 are most appropriate. While we may be able to give you extra space if necessary please try to consolidate the remaining figures, these can take up a full length page. Scale bars must be present on all microscopy images, including inset magnifications.

3) Statistical analysis: Error bars on graphic representations of numerical data must be clearly described in the figure legend. The number of independent data points (n) represented in a graph must be indicated in the legend. Statistical methods should be explained in full in the materials and methods. For figures presenting pooled data the statistical measure should be defined in the figure legends. Please also be sure to indicate the statistical tests used in each of your experiments (both in the figure legend itself and in a separate methods section) as well as the parameters of the test (for example, if you ran a t-test, please indicate if it was one- or two-sided, etc.). Also, if you used parametric tests, please indicate if the data distribution was tested for normality (and if so, how). If not, you must state something to the effect that "Data distribution was assumed to be normal but this was not formally tested."

4) Title: The title should be accessible to a general readership but concise and less than 100 characters including spaces. Your title exceeds this limit. We therefore suggest the following title: "Single-molecule localization microscopy unravels the kinetochore nanostructure in fission yeast".

5) Materials and methods: Should be comprehensive and not simply reference a previous publication for details on how an experiment was performed. Please provide full descriptions (at least in brief) in the text for readers who may not have access to referenced manuscripts. The text should not refer to methods "...as previously described." JCB formatting does not allow for supplementary methods or text, please move this section to the main methods section.

6) For all cell lines, vectors, constructs/cDNAs, etc. - all genetic material: please include database / vendor ID (e.g., Addgene, ATCC, etc.) or if unavailable, please briefly describe their basic genetic features, even if described in other published work or gifted to you by other investigators (and provide references where appropriate). Please be sure to provide the sequences for all of your oligos: primers, si/shRNA, RNAi, gRNAs, etc. in the materials and methods. You must also indicate in the methods the source, species, and catalog numbers/vendor identifiers (where appropriate) for all of your antibodies, including secondary. If antibodies are not commercial, please add a reference citation if possible.

7) Microscope image acquisition: The following information must be provided about the acquisition and processing of images:

- a. Make and model of microscope
- b. Type, magnification, and numerical aperture of the objective lenses
- c. Temperature
- d. Imaging medium
- e. Fluorochromes
- f. Camera make and model
- g. Acquisition software
- h. Any software used for image processing subsequent to data acquisition. Please include details and types of operations involved (e.g., type of deconvolution, 3D reconstitutions, surface or volume rendering, gamma adjustments, etc.).

8) References: There is no limit to the number of references cited in a manuscript. References should be cited parenthetically in the text by author and year of publication. Abbreviate the names of journals according to PubMed. JCB formatting does not allow for supplementary references, please remove these and add any non-duplicate references to the main reference list.

9) Supplemental materials: As mentioned above we can give you more room for supplemental figures but please try to consolidate these as much as possible. Please also note that tables, like figures, should be provided as individual, editable files. A summary of all supplemental material should appear at the end of the Materials and methods section. Please include one brief sentence per item.

10) eTOC summary: A ~40-50 word summary that describes the context and significance of the findings for a general readership should be included on the title page. The statement should be written in the present tense and refer to the work in the third person. It should begin with "First author name(s) et al..." to match our preferred style.

11) Conflict of interest statement: JCB requires inclusion of a statement in the acknowledgements regarding competing financial interests. If no competing financial interests exist, please include the following statement: "The authors declare no competing financial interests." If competing interests are declared, please follow your statement of these competing interests with the following statement: "The authors declare no further competing financial interests."

12) A separate author contribution section is required following the Acknowledgments in all research manuscripts. All authors should be mentioned and designated by their first and middle initials and full surnames. We encourage use of the CRediT nomenclature (<https://casrai.org/credit/>).

13) ORCID IDs: ORCID IDs are unique identifiers allowing researchers to create a record of their various scholarly contributions in a single place. At resubmission of your final files, please consider providing an ORCID ID for as many contributing authors as possible.

B. FINAL FILES:

Thank you for this interesting contribution, we look forward to publishing your paper in Journal of Cell Biology.

Sincerely,

Arshad Desai, PhD
Monitoring Editor
Journal of Cell Biology

Dan Simon, PhD
Scientific Editor
Journal of Cell Biology

Reviewer #1 (Comments to the Authors (Required)):

Virant and colleagues use multi-color single molecule fluorescence microscopy to define the nanoscale organization and stoichiometry of key kinetochore proteins in metaphase fission yeast cells. This work provides an important bridge between earlier studies of kinetochore architecture using fluorescence microscopy and structural biology of individual kinetochore proteins. Together with the companion paper by Lando and colleagues that presents a similar investigation of the budding yeast kinetochore, this study represents a major contribution to fully understanding the architecture of one of the more complex organelles in eukaryotic cells.

The authors have fully addressed my comments. I have only one minor comment for the revising the manuscript.

It is assumed here that the centroids of a POI, the centromere (Cnp1), and the spindle pole body (Sad1) lie along a straight line representing the microtubule. For the sake of a complete discussion, the authors should cite work by McAinsh and Burroughs (eLife 2016) presenting evidence that the outer kinetochore proteins can 'swivel' about the centromere in human cells.

Reviewer #2 (Comments to the Authors (Required)):

The authors have addressed my initial concerns and I feel that this manuscript is appropriate for publication with the accompanying paper. Specifically, the authors have altered the sad1-scarlet-I images in Figure 1, they look much better. They have addressed my concern about the dimensions used. They have addressed the mitotic stage (metaphase/anaphase) more clearly and have clarified a number of other queries I had concerning measurements of POI copy numbers and distances. The manuscript and analysis are greatly improved.

Point-to-Point responses below!

Dear Prof. Endesfelder,

Thank you for submitting your revised manuscript entitled "Unraveling the kinetochore nanostructure in *Schizosaccharomyces pombe* using multi-color single-molecule localization microscopy." We would be happy to publish your paper in JCB pending the minor textual changes recommended by the reviewers as well as final revisions necessary to meet our formatting guidelines (see details below).

Thanks a lot, our responses can be found below!

A. MANUSCRIPT ORGANIZATION AND FORMATTING:

Full guidelines are available on our Instructions for Authors page, <https://jcb.rupress.org/submission-guidelines#revised>. ****Submission of a paper that does not conform to JCB guidelines will delay the acceptance of your manuscript.****

1) Text limits: Character count for Reports is < 20,000, not including spaces and there should be a single combined 'Results and Discussion' section. Count includes title page, abstract, introduction, results, discussion, and acknowledgments. Count does not include materials and methods, figure legends, references, tables, or supplemental legends.

We have now put the sections in the correct order and shortened the manuscript to < 20,000 characters as required. Importantly, we have not changed the content, we have deleted filler words such as "already, also, additionally", abbreviated frequently used words such as "kinetochore" (KT) and converted the grammar to shorter versions where possible.

2) Figure formatting: Reports may have up to 5 main text figures and are generally limited to 3 supplemental figures. We will be able to give you extra space for supplemental materials but since you currently only have 2 main figures we ask that you please move some of the supplemental data to the main text. We think that Figures S7, S9, S10, and maybe S12 are most appropriate. While we may be able to give you extra space if necessary please try to consolidate the remaining figures, these can take up a full length page. Scale bars must be present on all microscopy images, including inset magnifications.

We have restructured the figures and believe that we have found a good version with three main figures and three supplementary figures. Importantly, none of the original graphic content has been omitted, it has just been rearranged and combined into less figures. We hope that this proposal fits JCB's formatting. We also carefully checked scale bars, info in captions etc. and believe that it now is complete.

3) Statistical analysis: Error bars on graphic representations of numerical data must be clearly described in the figure legend. The number of independent data points (n) represented in a graph must be indicated in the legend. Statistical methods should be explained in full in the materials and methods. For

figures presenting pooled data the statistical measure should be defined in the figure legends. Please also be sure to indicate the statistical tests used in each of your experiments (both in the figure legend itself and in a separate methods section) as well as the parameters of the test (for example, if you ran a t-test, please indicate if it was one- or two-sided, etc.). Also, if you used parametric tests, please indicate if the data distribution was tested for normality (and if so, how). If not, you must state something to the effect that "Data distribution was assumed to be normal but this was not formally tested."

Most of the requested statistical information was already present in the last version, but we have added info where it was missing.

4) Title: The title should be accessible to a general readership but concise and less than 100 characters including spaces. Your title exceeds this limit. We therefore suggest the following title: "Single-molecule localization microscopy unravels the kinetochore nanostructure in fission yeast".

We have discussed the title among the authors and would like to shorten it by using the short version of "Single-molecule localization microscopy", SMLM. So our title would be:

Unraveling the kinetochore nanostructure in *Schizosaccharomyces pombe* using multi-color SMLM imaging (exactly 100 characters)

Our reasoning is that this acronym is very well-known after almost 20 years of SMLM imaging and it is therefore fine to not write it in long, similar to "STED" for example, no one uses the long version "Stimulated Depletion Microscopy" anymore, also not in titles. It is just STED microscopy.

5) Materials and methods: Should be comprehensive and not simply reference a previous publication for details on how an experiment was performed. Please provide full descriptions (at least in brief) in the text for readers who may not have access to referenced manuscripts. The text should not refer to methods "...as previously described." JCB formatting does not allow for supplementary methods or text, please move this section to the main methods section.

We have carefully checked the M & M and adjusted our supplementary text, which is now part of M & M.

6) For all cell lines, vectors, constructs/cDNAs, etc. - all genetic material: please include database / vendor ID (e.g., Addgene, ATCC, etc.) or if unavailable, please briefly describe their basic genetic features, even if described in other published work or gifted to you by other investigators (and provide references where appropriate). Please be sure to provide the sequences for all of your oligos: primers, si/shRNA, RNAi, gRNAs, etc. in the materials and methods. You must also indicate in the methods the source, species, and catalog numbers/vendor identifiers (where appropriate) for all of your antibodies, including secondary. If antibodies are not commercial, please add a reference citation if possible.

We have carefully checked all this info.

7) Microscope image acquisition: The following information must be provided about the acquisition and processing of images:

- a. Make and model of microscope
- b. Type, magnification, and numerical aperture of the objective lenses

- c. Temperature
- d. Imaging medium
- e. Fluorochromes
- f. Camera make and model
- g. Acquisition software
- h. Any software used for image processing subsequent to data acquisition. Please include details and types of operations involved (e.g., type of deconvolution, 3D reconstitutions, surface or volume rendering, gamma adjustments, etc.).

We have carefully checked all this info and added the missing temperature information.

8) References: There is no limit to the number of references cited in a manuscript. References should be cited parenthetically in the text by author and year of publication. Abbreviate the names of journals according to PubMed. JCB formatting does not allow for supplementary references, please remove these and add any non-duplicate references to the main reference list.

We made sure that we used the JCB formatting (and removed/restructured the supplementary references).

9) Supplemental materials: As mentioned above we can give you more room for supplemental figures but please try to consolidate these as much as possible. Please also note that tables, like figures, should be provided as individual, editable files. A summary of all supplemental material should appear at the end of the Materials and methods section. Please include one brief sentence per item.

We added the summary at the end of M & M as requested and drastically lowered the number of SI pieces.

10) eTOC summary: A ~40-50 word summary that describes the context and significance of the findings for a general readership should be included on the title page. The statement should be written in the present tense and refer to the work in the third person. It should begin with "First author name(s) et al..." to match our preferred style.

Our suggestion for the eTOC is:

Virant et al. build an *in situ* model of the *Schizosaccharomyces pombe* kinetochore by determining inter-protein cluster distances and protein copy numbers. In addition to confirming its overall conserved nature as known from *in vitro* data, they point out *S. pombe* specifics, e.g. within DASHc and the inner kinetochore structure.

11) Conflict of interest statement: JCB requires inclusion of a statement in the acknowledgements regarding competing financial interests. If no competing financial interests exist, please include the following statement: "The authors declare no competing financial interests." If competing interests are declared, please follow your statement of these competing interests with the following statement: "The authors declare no further competing financial interests."

Added the statement

12) A separate author contribution section is required following the Acknowledgments in all research manuscripts. All authors should be mentioned and designated by their first and middle initials and full surnames. We encourage use of the CRediT nomenclature (<https://casrai.org/credit/>).

Changed the names into the required format.

13) ORCID IDs: ORCID IDs are unique identifiers allowing researchers to create a record of their various scholarly contributions in a single place. At resubmission of your final files, please consider providing an ORCID ID for as many contributing authors as possible.

Added ORCID identifiers.

B. FINAL FILES:

Thank you for this interesting contribution, we look forward to publishing your paper in Journal of Cell Biology.

Sincerely,

Arshad Desai, PhD
Monitoring Editor
Journal of Cell Biology

Dan Simon, PhD
Scientific Editor
Journal of Cell Biology

Reviewer #1 (Comments to the Authors (Required)):

Virant and colleagues use multi-color single molecule fluorescence microscopy to define the nanoscale organization and stoichiometry of key kinetochore proteins in metaphase fission yeast cells. This work provides an important bridge between earlier studies of kinetochore architecture using fluorescence microscopy and structural biology of individual kinetochore proteins. Together with the companion paper by Lando and colleagues that presents a similar investigation of the budding yeast kinetochore, this study represents a major contribution to fully understanding the architecture of one of the more complex organelles in eukaryotic cells.

The authors have fully addressed my comments. I have only one minor comment for the revising the manuscript.

Thanks a lot!

It is assumed here that the centroids of a POI, the centromere (Cnp1), and the spindle pole body (Sad1) lie along a straight line representing the microtubule. For the sake of a complete discussion, the authors should cite work by McAinsh and Burroughs (eLife 2016) presenting evidence that the outer kinetochore proteins can 'swivel' about the centromere in human cells.

We added the contents and the reference into the text.

Reviewer #2 (Comments to the Authors (Required)):

The authors have addressed my initial concerns and I feel that this manuscript is appropriate for publication with the accompanying paper. Specifically, the authors have altered the sad1-scarlet-I images in Figure 1, they look much better. They have addressed my concern about the dimensions used. They have addressed the mitotic stage (metaphase/anaphase) more clearly and have clarified a number of other queries I had concerning measurements of POI copy numbers and distances. The manuscript and analysis are greatly improved.

Thanks a lot!